# Task-Related Token Compression in Multimodal Large Language Models from an Explainability Perspective

**Lei Lei**[1,3*]    **Jie Gu**[2†]    **Xiaokang Ma**[2]    **Chu Tang**[2]    **Jingmin Chen**[2]    **Tong Xu**[1†]
[1] University of Science and Technology of China    [2] Rightly Robotics
[3] Shanghai Innovation Institute
{lily168,tongxu}@ustc.edu.cn;{jgu,xma,chu.tang,jingmin.chen}@rightly.ai

## Abstract

Existing Multimodal Large Language Models (MLLMs) process a large number of visual tokens, leading to significant computational costs and inefficiency. Instruction-related visual token compression demonstrates strong task relevance, which aligns well with MLLMs' ultimate goal of instruction following. Previous works generally assume that visual tokens achieve better vision–language alignment in the shallow layers of LLMs, which have led to task-related token compression being primarily applied in intermediate LLM layers. In contrast, our study reveals that with proper selection, task-related token compression is feasible at the input stage of LLM with negligible performance loss. This new paradigm significantly reduces task-irrelevant visual tokens and its model-agnostic design enables application without modifying the LLM architecture. Specifically, we suggest that explainability methods for transformer-based architechtures can evaluate the global importance of each visual token with respect to the given instruction, which can effectively guide the task-related token compression for MLLMs. Furthermore, we propose to learn a mapping from the attention map of the first LLM layer to the explanation results, thereby avoiding the need for a full inference pass. Interestingly, this mapping can be learned using a simple and lightweight convolutional network, whose training is efficient and independent of MLLMs. Extensive experiments on 13 image and video benchmarks across three leading MLLMs (Qwen2-VL, LLaVA-OneVision, and VILA1.5) demonstrate the remarkable effectiveness and strong generalization of our approach. Additionally, our new compression paradigm achieves faster inference with reductions in both prefilling time and KV-cache memory.

## 1 Introduction

With large language models (LLMs) providing a strong foundation Brown et al. (2020); OpenAI (2023); Touvron et al. (2023); Bi et al. (2024), research on multimodal large language models (MLLMs) has gained significant momentum Liu et al. (2023); Chen et al. (2023); Zhu et al. (2024); Bai et al. (2023). Considerable progress has been achieved in various image- and video-related tasks Chen et al. (2024d); Anil et al. (2023). A common paradigm among existing MLLMs is to jointly feed visual tokens (generated by a vision encoder) and textual tokens into the LLM for cross-modal alignment and integration Liu et al. (2023); Zhu et al. (2024); Li et al. (2023b). This paradigm introduces substantial memory and computational overhead due to the high volume of visual tokens, which grows rapidly with higher resolutions or frame rates Wang et al. (2024b); Zhang et al. (2024a). Consequently, there is a pressing need for effective token compression techniques.

Previous exploration of visual token compression methods can be roughly divided into two categories. The first aims to obtain more compact and fewer visual representations (especially for videos) in a task/instruction-agnostic manner (independent of LLM) Bolya et al. (2023); Yang et al.

---

*Work done during interships in Rightly Robotics.
†Corresponding author.

(2024); Shen et al. (2025b); Wang et al. (2025); Shen et al. (2025a). We argue that visual representations are an integral part of MLLMs and serve as the foundation for achieving strong performance and generalization. Many state-of-the-art(SOTA) MLLMs already incorporate built-in, task-agnostic compression mechanisms (e.g., spatial and temporal pooling) instead of relying on separate compression techniques applied afterward Wang et al. (2024b;a); Li et al. (2024c). The second category focuses on selecting visual tokens that are most relevant to the given instruction. FastV Chen et al. (2024b) is a pioneering work that highlights the importance of retaining all shallow-layer visual tokens in LLMs for lossless compression. While this assumption has been adopted by many subsequent studies Zhang et al. (2024b); Zhao et al. (2024); Xing et al. (2024); Tan et al. (2025); Huang et al. (2024); Wen et al. (2025), we believe it remains open to question: *Are all visual tokens in the shallow layers of LLM truly indispensable for task-related compression?*

This paper seeks to answer the question of whether an effective task-related token compression approach prior to the LLM exists but remains undiscovered, or whether it is inherently infeasible. To the best of our knowledge, our work is among the earliest efforts to investigate this issue. We first explore the use of explainability methods to assess visual token importance with respect to the instruction. Explainability methods for transformer-based architecture generally iteratively update a relevance map across layers using gradient-weighted multi-head attentions Chefer et al. (2021b;a), which effectively captures the global relevance scores of visual tokens. Relevance scores indicating the contributions of input tokens to output can be used to rank and prune less important visual tokens for compression. Comprehensive experiments conducted on both image and video data across three representative MLLMs demonstrate the effectiveness of such a compressor. The results indicate that, with proper selection, pruning tokens that are not relevant to the task at the LLM input stage is indeed feasible. Moreover, unlike previous works motivated by observations derived from specific network architectures (*e.g.*, LLaVA) Chen et al. (2024b); Tan et al. (2025), which limits their generality and transferability, our explainability-based approach is broadly applicable. Rather than relying on the behaviors of specific models, it leverages the inherent characteristics of the applied model.

After validating that the explanation results are effective compression indicators, a lightweight model capable of generating an alternative to the relevance map is further needed to enable efficient and practical deployment. Interestingly, this goal can be achieved by training a simple fully convolutional network that predicts relevance based on the first-layer attention map of the LLM. The training process is highly efficient (*e.g.*, training a 5-layer network using only 10K image data) and does not involve any changes to the MLLM itself. Using the predicted relevance, token compression can be performed prior to the prefill phase with negligible extra computational cost. As a result, both computational and memory overhead during inference are significantly reduced, with no modifications required to either the prefill or decode phases. Last but not least, our approach generalizes well across various architectures, benefiting from the broadly applicable nature of explainability methods and the MLLM-agnostic design of the auxiliary training.

To thoroughly assess the capability of our approach, we apply it to three prominent models with different architectures and visual representations: VILA1.5, LLaVA-OneVision, and Qwen2-VL. We include 13 widely used image and video benchmarks that span a wide range of visual complexities and tasks, ensuring a comprehensive evaluation. Notably, our method achieves significant compression by pruning 75% of video tokens while retaining more than 97% of the original performance across all benchmarks for both VILA1.5 and LLaVA-OneVision. It also performs well on image tasks, where up to 50% of image tokens can be removed with only a minimal performance drop: maintaining over 96% of baseline performance for Qwen2-VL and LLaVA-OneVision.

In summary, the contributions of the work are threefold: (i) reveal that explainability methods can well evaluate the importance of visual tokens, enabling effective token compression. (ii) propose a highly efficient token compressor by learning from explanation results. It allows token compression to be performed before the LLM, significantly reducing inference costs at both the prefill and decode phases. (iii) Validate the effectiveness and generalization of our method through extensive experiments on a wide range of image and video benchmarks across different leading MLLMs.

## 2 RELATED WORK

**Multimodal Large Language Models.** Benefiting from advancements in large language models (LLMs) OpenAI (2023); Touvron et al. (2023); Bi et al. (2024), multimodal large language models

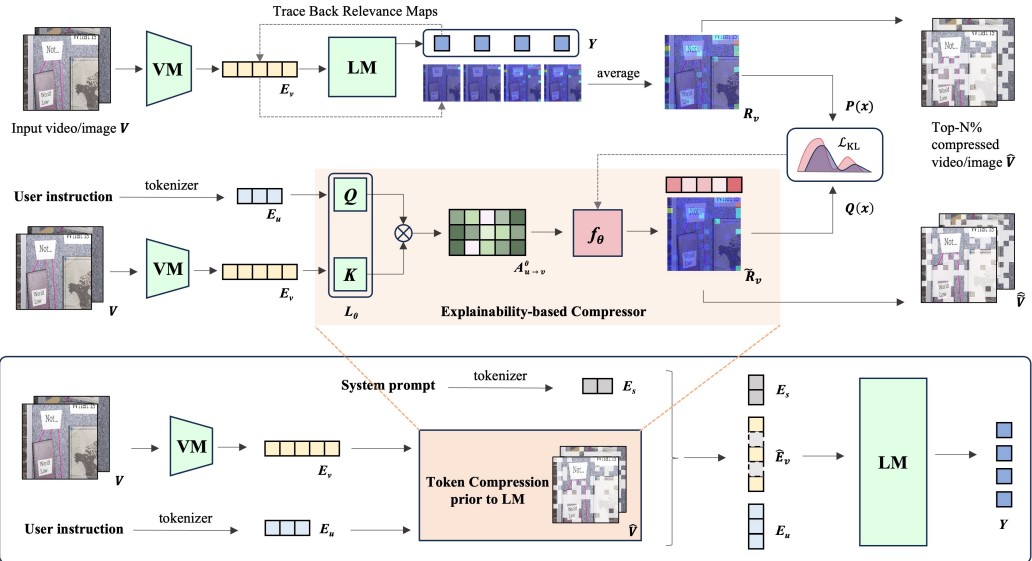

Figure 1: **Overview of our method.** The top portion illustrates the details of our explainability-based compression approach: an explainability method can reveal the important visual tokens (first row, Section 3.2); a lightweight model can then be trained to approximate this explainability and serve as a compression indicator (second row, Section 3.3). The bottom portion shows a general inference framework for MLLMs, where the resulting compressor is applied at the LLM input stage.

(MLLMs) have gained considerable attention due to the powerful ability in multi-modal understanding and reasoning Liu et al. (2023); Chen et al. (2023); Bai et al. (2023); Chen et al. (2024d); Anil et al. (2023). Recent advances Li et al. (2024a); Wang et al. (2024b); Zhang et al. (2024a) tend to handle images with higher resolution and videos with more frames, which significantly increases the number of visual tokens and thus the computational burden. This reveals the necessity for token compression strategies that can balance efficiency and effectiveness. Our work proposes a new token compression paradigm, which removes task-irrelevant visual tokens at the LLM input stage, significantly reducing computational costs without sacrificing performance.

**Visual Token Compression.** Existing visual token compression methods for MLLMs can be broadly categorized into: task/instruction-agnostic compression Bolya et al. (2023); Yang et al. (2024); Shen et al. (2025b); Alvar et al. (2025) and task/instruction-related compression Chen et al. (2024b); Xing et al. (2024); Huang et al. (2024); Wen et al. (2025). The first category of methods typically introduces additional modules to merge redundant visual tokens based on similarity, addressing the limitations of existing models. However, many recent works have developed techniques to obtain more compact visual representations when building MLLMs Wang et al. (2024b); Chen et al. (2024a). Notably, task/instruction-related compression can further reduce the number of visual tokens on models already incorporate built-in compression mechanisms, offering greater potential for efficiency gains. FastV Chen et al. (2024b) represents a typical method of the second category, which rely on shallow-layer attention maps of the LLM for compression. In this work, we explores the feasibility of an effective task-related token compression prior to the LLM, which functions independently of the architecture and can be applied broadly across different MLLMs.

## 3 METHOD

### 3.1 BACKGROUND AND MOTIVATION

Current Multimodal Large Language Models (MLLMs) typically follow a framework in which a vision encoder is incorporated to encode visual signals into a sequence of tokens Liu et al. (2023); Bai et al. (2023); Chen et al. (2023). Specifically, multiple frames or patches are sampled from a video or an image, and their corresponding visual tokens are encoded. These visual tokens are then flattened and concatenated with textual prompt tokens before being fed into a Large Language Model (LLM) to generate a response. Formally, let $V$ be the video or image, and let $\mathbf{VM}$ and $\mathbf{LM}$ represent the

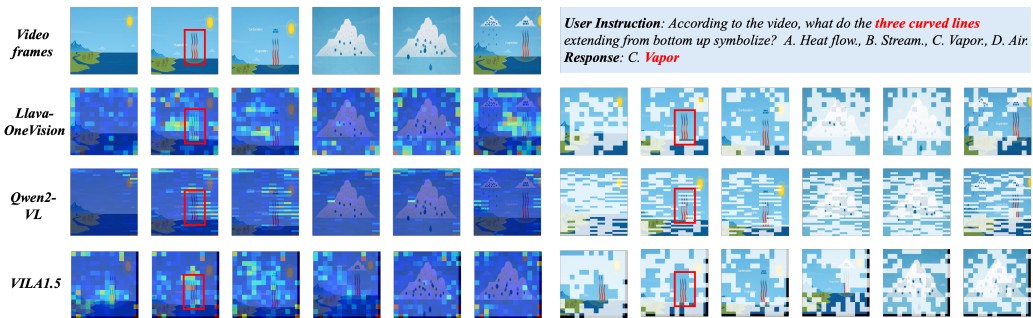

Figure 2: **Visualization of $R_v$ obtained via the explainability method (left) and the corresponding token pruning results (right).** Based on $R_v$, the top 50% of visual tokens are retained and the rest masked in white. Given the instruction querying the "three curved lines", $R_v$ highlights the regions corresponding to the "three curved lines", guiding the selective retention of the associated visual tokens. All three MLLMs generate the correct answer using only the retained tokens. More visualization cases are presented in Appendix B.

vision encoder and the language model, respectively. The visual token embeddings $E_v$ can be represented as $E_v = \mathbf{VM}(V) \in \mathbb{R}^{N_v \times C}$, where $N_v$ is the number of visual tokens and $C$ is the feature dimension. [1] Let $E_s \in \mathbb{R}^{N_s \times C}$ and $E_u \in \mathbb{R}^{N_u \times C}$ denote the token embeddings of the system prompt and user instruction, respectively. By feeding $E_v$ together with $E_s$ and $E_u$ into the LLM, a textual response is generated, *i.e.*, $Y = \mathbf{LM}(E_s, E_v, E_u)$. An additional compression module $\mathbf{Comp}$ can be introduced to prune visual tokens, while keeping the MLLM architecture—including both the $\mathbf{VM}$ and the $\mathbf{LM}$—unchanged during this pruning process.

$E_v$ can be considered as general-purpose representations of visual signals that are task/instruction-agnostic. Recent advances have already incorporated built-in compression mechanisms (e.g., spatial and temporal pooling) to reduce task-agnostic redundancy, enabling the construction of more compact $E_v$ when building MLLMs Wang et al. (2024b;a); Li et al. (2024c). Rather than applying another round of task-agnostic compression to $E_v$ (as in methods Bolya et al. (2023); Shen et al. (2025a)), our objective is to further address task-related redundancy by evaluating the importance of each token in $E_v$ with respect to a given instruction and selectively pruning those that contribute less. Recent advances have developed techniques to reduce the number of visual tokens to obtain a more compact $E_v$ when building MLLMs Wang et al. (2024b;a); Li et al. (2024c). Therefore, instead of further compressing $E_v$ in isolation (namely task-agnostic compresssion methods like Bolya et al. (2023); Shen et al. (2025a)), our objective is to assess the importance of each token in $E_v$ with respect to a given instruction, and subsequently prune those that are less essential. Moreover, we investigate how to perform token compression prior to LLM computation, *i.e.*, compressing $E_v$ to $\hat{E}_v = \mathbf{Comp}(E_v|E_u) \in \mathbb{R}^{\hat{N}_v \times C}$ and then computing $Y = \mathbf{LM}(E_s, \hat{E}_v, E_u)$, where $\hat{N}_v$ is much smaller than $N_v$. In contrast to previous methods Chen et al. (2024b); Huang et al. (2024), our method does not require any modifications to the prefill and decode phases during inference, and computational and memory overhead can be significantly reduced in both phases.

The details of our approach are presented below. In Section 3.2, we introduce explainability methods to assess the importance of visual tokens and guide token compression. A learning mechanism is then proposed to predict the explanation results in Section 3.3, which ultimately enables effective token compression at the LLM input stage.

## 3.2 TOKEN COMPRESSION WITH EXPLAINABILITY

To reduce task-related redundancy at the token level, we need to estimate the contribution of each visual token to the model response. Explainability methods for LLMs facilitate this goal by generating a relevance map through the integration of attention weights and corresponding gradients, effectively revealing where the model genuinely focuses. The resulting relevance map highlights the

---

[1]A cross-modal projector is commonly employed in such architectures. For notational simplicity, we denote both the vision encoder and the projector by $\mathbf{VM}$.

contributions, enabling us to rank and prune these visual tokens accordingly. The pipeline for this section is shown in the first row of Figure 1.

**Relevance Maps by Explainability Method.** We adopt a generic explainability method similar to Yao et al. (2024); Chefer et al. (2021b) to compute the relevance of the response-to-vision. The relevance values reveal the distribution of importance across visual tokens utilized by the LLM. Without loss of generality, assume that the LLM in an MLLM has $L$ layers, and denote the generated sequence of textual tokens as $Y = \{y_0, y_1, \ldots, y_{T-1}\}$. Specifically, we trace back the semantic relevance flow from generated tokens to raw visual inputs. For each $y_t$ at the $t$-th generation step, the relevance map $R_t$ is first initialized as an identity matrix and then iteratively updated across layers. Denote $A_t^l$ and $\nabla A_t^l$ as the multi-head attention map and the corresponding gradients in the $l$-th layer, obtained during the forward and backward passes, respectively. $R_t$ is updated as:

$$R_t = R_t + \mathbb{E}_h((A_t^l \odot \nabla A_t^l)^+) \cdot R_t, \tag{1}$$

where $\odot$ represents Hadamard product, and $\mathbb{E}_h$ is the mean across the heads dimension. The update is performed from the 0-th layer to the last layer. In the end, the relevance of $y_t$ to visual signals can be extracted by indexing the corresponding positions in the last row of $R_t$, that is, $R_t[-1, N_s : N_s + N_v]$. Finally, we aggregate visual relevance across all time steps $t$ by averaging, obtaining the overall visual relevance scores $R_v \in \mathbb{R}^{1 \times N_v}$ with respect to the current response. A detailed analysis of Eq. 1 is provided in Appendix A. This well-grounded importance assessment $R_v$ can then be used to rank and select visual tokens.

**Visual Token Compression Using Relevance Scores.** The importance of visual tokens related to the instruction can be ranked according to $R_v$. We can prune the less important visual tokens down to a target count of $\hat{N}_v$, resulting in compressed token embeddings $\hat{E}_v$ as LLM input.

**Observation.** We visualize $R_v$ and the corresponding token pruning results for LLaVA-OneVision, Qwen2-VL, and VILA1.5 in Figure 2. While visualizations from different MLLMs show varying appearances due to differences in how each model processes visual input, they exhibit clear common patterns — $R_v$ for each model consistently highlights the regions corresponding to the "three curved lines" in the video, demonstrating the robustness of our method. Moreover, experimental results show that retaining 50% of the original visual tokens based on $R_v$ preserves over 98% of the performance on image benchmarks and 99% on video benchmarks (see Section 4.2 for details). We draw the following conclusion: *the explanation results faithfully capture the visual information essential for the MLLM to answer the question, and retaining only the corresponding visual tokens does not compromise model performance.*

### 3.3 Explainability-based Compressor Learning

The relevance map offers valuable insights into achieving token compression at the LLM input level. However, its practical application is limited by the fact that $R_v$ is derived post-hoc – only after the model has already generated the output. To address this limitation, we propose to approximate $R_v$ using a standalone module trained independently of the MLLM. By learning to capture attention patterns and generate relevance estimates $\tilde{R}_v$, this module enables token compression prior to LLM without modifying or retraining the MLLM. Importantly, this module prioritizes efficiency: it is lightweight, requires a small amout of training data, and can be trained quickly, making it practically applicable. The pipeline for this section is shown in the second row of Figure 1.

**Model Architecture.** As shown in Eq. 1, the relevance map is essentially obtained by aggregating attention maps, suggesting that learning a mapping from attention maps to relevance maps is promising. Interestingly yet reasonably, we find in practice that a simple convolutional network applied to the first-layer attention suffices, which guarantees the compressor's efficiency in terms of model size, training time, and computation (implementation details and efficiency analysis can be found in Appendix C and Appendix D). Formally, let $A^0$ be the first-layer attention map. Similar to Chen et al. (2024b); Zhao et al. (2024), we focus specifically on the attention scores that visual tokens receive from textual instruction tokens. Accordingly, we extract the submap $A_{u \to v}^0 \in \mathbb{R}^{N_u \times N_v}$ by indexing the corresponding positions. We then average the $N_u$ scores for each visual token to obtain a compact representation, resulting in $A_v^0 \in \mathbb{R}^{1 \times N_v}$.[2] This averaged attention vector $A_v^0$ is

---

[2]We omit the head dimension for notational simplicity.

subsequently fed into a 1D convolutional model $f_\theta$ to predict visual relevance:

$$\tilde{R}_v = f_\theta(A_v^0). \tag{2}$$

Note that a softmax operation is applied at the end of $f_\theta$, making $\tilde{R}_v$ a probability distribution. In addition, a separate instance of $f_\theta$ is used for each MLLM, because it is trained to approximate the explainability patterns specific to that particular MLLM.

**Training Objectives.** We process $R_v$ into the training label $R_v^*$ by masking the bottom 50% values Gu et al. (2021) and normalizing the remainder into a probability distribution. Instead of softmax—which yields near-uniform values due to the closeness of raw scores—we normalize by dividing each score by the total, thereby preserving relative differences. Finally, given $R_v^*$ and $\tilde{R}_v$, the Kullback–Leibler (KL) divergence is used to measure the difference, defining the loss function:

$$\mathcal{L}_{KL} = \mathbf{KL}(R_v^*||\tilde{R}_v). \tag{3}$$

**Oberservation.** The learned $f_\theta$ can be seamlessly integrated into the MLLM inference pipeline to generate $\tilde{R}_v$, which can guide the token compression. As shown in Figure 1, a visualization of $R_v$ and $\tilde{R}_v$ is given in the first and second rows, along with their corresponding pruning results, respectively. One can see that $\tilde{R}_v$ closely resembles $R_v$. Important visual regions related to the question are highlighted in both maps. This observation provides evidence that the lightweight model $f_\theta$ can indeed be efficiently and effectively trained to approximate $R_v$, allowing lossless token compression at the LLM input stage. Quantitative experimental results further support this conclusion (see Section 4.3 for details).

## 4 EXPERIMENTS

### 4.1 EXPERIMENTAL SETUP

**Models.** Experiments are conducted on three MLLMs with different architectures for extensive validation, *i.e.*, LLaVA-OneVision-7B Li et al. (2024a), Qwen2-VL-7B Wang et al. (2024b) and VILA1.5-8B Liu et al. (2024c). These models exemplify recent advances in handling high-resolution and long visual inputs. LLaVA-OneVision and Qwen2-VL support arbitrary resolution/length, with Qwen2-VL further introducing dynamic resolution and token aggregation for compact visual representations. VILA1.5 applies spatial token compression when processing images or video frames. They thus provide a strong basis for evaluating our task-related compression, which reduces instruction-related redundancy beyond their built-in instruction-agnostic compression.

**Evaluation Tasks.** We thoroughly evaluate our method on 13 widely used image and video benchmarks. For image tasks, MME Fu et al. (2023) (all-round capability), MMStar Chen et al. (2024c) (data contamination), MMVet Yu et al. (2024) (subjective evaluation), SEED-Bench Li et al. (2023a) (all-round capability), POPE Li et al. (2023c) (hallucination evaluation), TextVQA Singh et al. (2019) (OCR reasoning), and MMBench Liu et al. (2024a) (all-round capability) are included, covering various aspects of MLLM performance. For video evaluation, we select Video-MME(wo sub.) Fu et al. (2024), MVBench Li et al. (2024b), MMBench-Video Fang et al. (2024), NExT-QA Xiao et al. (2021), and ActivityNetQA Yu et al. (2019), providing comprehensive coverage of video understanding abilities across different tasks and video durations. For comparison, we take existing SOTA task-related token compression methods such as FastV Chen et al. (2024b), Pyramid-Drop Xing et al. (2024), Dart Wen et al. (2025) as the primary baselines, which perform compression in the LLM intermediate layers on visual tokens fused with textual information in the shallow layers.

**Implementation Details.** *Training $f_\theta$.* $f_\theta$ is implemented as a five-layer convolutional network, with each layer employing a 1D depthwise separable convolution Chollet (2017). The training data of $f_\theta$ is collected from general-domain high-quality datasets: a subset of LLaVA-Video Zhang et al. (2024c) for videos and a subset of Infinity-MM Gu et al. (2024) for images. To ensure high diversity, we adopt a sampling strategy that covers a wide range of task types and video duration. Implementation details including the sampling strategy of training data, the generation of $R_v$, and the training procedure of $f_\theta$ are provided in Appendix C.

*Inference*. We conducted all experiments on A100 GPUs (80GB) and used VLMEvalKit Duan et al. (2024) for benchmarking. Details of the inference settings can be found in Appendix C.

Table 1: **The relevance $R_v$ effectively guides token compression under different retention ratios.** Avg. means the average of performance preservation ratios across all image/video benmarks.

| Method | Retention Ratio | Image Benchmark | | | Avg.(%) | Video Benchmark | | | Avg.(%) |
|---|---|---|---|---|---|---|---|---|---|
| | | MME | MMStar | MMVet | | Video-MME | MVBench | MMBench-V | |
| LLaVA-OneVision | 100% | 1997.7 | 60.5 | 48.7 | 100 | 53.6 | 41.2 | 0.41 | 100 |
| LLaVA-OneVision | 50% | 1974.2 | 59.7 | 47.2 | **98.1** | 54.3 | 41.1 | 0.40 | **99.5** |
| w/GAE-Based Compressor | 25% | 1977.3 | 59.3 | 47.0 | **97.8** | 53.8 | 40.9 | 0.40 | **99.1** |
| Qwen2-VL | 100% | 2295.1 | 60.4 | 54.0 | 100 | 50.4 | 51.0 | 1.23 | 100 |
| Qwen2-VL | 50% | 2297.1 | 60.3 | 53.2 | **99.5** | 51.0 | 50.7 | 1.19 | **99.1** |
| w/GAE-Based Compressor | 25% | 2299.1 | 58.7 | 51.7 | **97.7** | 50.3 | 49.7 | 1.17 | **97.5** |
| VILA1.5 | 100% | 1700.3 | 38.7 | 39.3 | 100 | 47.3 | 34.0 | 1.29 | 100 |
| VILA1.5 | 50% | 1740.5 | 37.2 | 38.0 | **98.4** | 47.9 | 34.2 | 1.26 | **99.8** |
| w/GAE-Based Compressor | 25% | 1722.1 | 35.7 | 35.6 | **94.7** | 47.1 | 35.1 | 1.28 | **100.7** |

Table 2: **Compare explainability-based compressor on image benchmarks.** Values marked with * in Retention Ratio denote the average retention ratio across LLM layers due to multi-stage compression in PDrop.

| Method | Retention Ratio | FLOPs | MME | MMStar | MMVet | SEED | POPE | TextVQA | MMBench | Avg.(%) |
|---|---|---|---|---|---|---|---|---|---|---|
| LLaVA-OV | 100% | 1.00× | 1997.7 | 60.5 | 48.7 | 76.7 | 87.4 | 69.3 | 80.6 | 100 |
| LLaVA-OV w/ FastV | 50% | 0.51× | 679.2 | 42.7 | 28.8 | 60.1 | 10.8 | 54.4 | 47.7 | 56.0 |
| LLaVA-OV w/ Pdrop | 51%* | 0.51× | 1974.7 | 55.4 | 41.7 | 74.8 | **87.0** | 64.9 | 79.9 | 95.1 |
| LLaVA-OV w/ Dart | 50% | 0.51× | 1977.5 | 55.2 | 42.3 | 74.3 | 85.8 | 65.4 | 79.2 | 95.0 |
| **LLaVA-OV w/ Ours** | 50% | **0.48×** | **1980.8** | **57.5** | **46.2** | **75.3** | 86.2 | **66.8** | **80.1** | **97.4** |
| LLaVA-OV w/ FastV | 25% | 0.27× | 527.7 | 41.5 | 20.6 | 56.1 | 10.6 | 38.4 | 42.8 | 47.3 |
| LLaVA-OV w/ PDrop | 25%* | 0.25× | 1888.3 | 50.1 | 34.7 | 70.4 | 79.6 | 54.8 | 75.7 | 86.3 |
| LLaVA-OV w/ Dart | 25% | 0.27× | 1905.0 | 48.7 | 36.7 | 69.5 | 80.9 | 57.8 | 76.2 | 87.5 |
| **LLaVA-OV w/ Ours** | 25% | **0.24×** | **1965.9** | **52.1** | 41.8 | **72.7** | **81.3** | **62.3** | **78.0** | **92.1** |
| Qwen2-VL | 100% | 1.00× | 2295.1 | 60.4 | 54.0 | 75.8 | 87.5 | 84.1 | 81.0 | 100 |
| Qwen2-VL w/ FastV | 50% | 0.51× | 1489.3 | 41.4 | 34.4 | 56.0 | 83.0 | 66.7 | 42.1 | 71.0 |
| Qwen2-VL w/ PDrop | 51%* | 0.51× | 2288.1 | 55.4 | 46.3 | 73.0 | 86.3 | 80.9 | 79.7 | 95.2 |
| Qwen2-VL w/ Dart | 50% | 0.51× | **2290.0** | 55.5 | 49.4 | 72.4 | **86.6** | **82.7** | 80.1 | 96.4 |
| **Qwen2-VL w/ Ours** | 50% | **0.49×** | 2288.3 | **55.9** | **51.9** | **73.2** | 86.4 | 82.6 | **80.9** | **97.4** |
| Qwen2-VL w/ FastV | 25% | 0.27× | 1415.3 | 37.6 | 31.4 | 51.4 | 77.6 | 63.2 | 39.1 | 66.0 |
| Qwen2-VL w/ PDrop | 25%* | 0.25× | 2216.3 | 51.1 | 42.3 | 67.4 | 83.2 | 75.4 | 77.0 | 89.7 |
| Qwen2-VL w/ Dart | 25% | 0.27× | 2184.5 | 51.3 | 45.6 | 68.2 | 84.3 | 76.7 | 77.5 | 91.1 |
| **Qwen2-VL w/ Ours** | 25% | **0.24×** | **2280.9** | **51.8** | **47.3** | 67.9 | **84.8** | **80.0** | **77.7** | **92.9** |

Following prior works Chen et al. (2024b); Ye et al. (2025), we report FLOPs as the primary metric for evaluating inference efficiency. For a fair comparison, we configure baselines for comparable FLOPs (e.g., pruning at the 2nd layer for FastV). Our method achieves a significant reduction in inference cost with only negligible additional computation. We provide a comprehensive analysis of FLOPs of our method, please refer to Appendix D.

## 4.2 Effectiveness of Compression with Explainability

We conduct experiments to verify whether the explanation results can guide token compression, *i.e.*, compressing $E_v$ to $\hat{E}_v$ according to $R_v$ and then feeding $\hat{E}_v$ into **LM** to generate a response. To assess effectiveness and generalization, we apply the method to three state-of-the-art MLLMs and test them on three image and three video benchmarks.

Table 1 reports the results under retention ratios of 50% and 25%. The strong performance across multiple models and benchmarks demonstrates the effectiveness and broad applicability of such an explainability-based token compressor. For Qwen2-VL, reducing visual tokens by 50% still preserves over 99% of the original performance on both image and video tasks. LLaVA-OneVision retains 99.1% of its video performance even with only 25% of tokens. VILA reduces visual tokens to 98 per image or frame at 50% retention, while maintaining 98% of the original image performance and nearly unchanged video performance. These observations indicate that token compression based on relevance $R_v$ effectively preserves the visual tokens essential for MLLMs to answer the question.

Table 3: **Compare explainability-based compressor on video benchmarks.** As videos generally exhibit greater visual redundancy, we also evaluate a lower retention ratio 10% to further assess compression robustness, with detailed results reported in the Appendix F.

| Method | Retention Ratio | FLOPs | Video-MME | MVBench | MMBench-Video | Next-QA multi-choice | Next-QA open-ended | Activity-QA | Avg.(%) |
|---|---|---|---|---|---|---|---|---|---|
| LLaVA-OV | 100% | 1.00× | 53.6 | 41.2 | 0.41 | 79.2 | 49.0 | 56.9 | 100 |
| LLaVA-OV w/ FastV | 50% | 0.48× | 42.3 | 25.0 | 0.35 | 66.5 | 36.0 | 48.5 | 78.0 |
| LLaVA-OV w/ PDrop | 50%* | 0.47× | 52.7 | 40.2 | 0.36 | 78.4 | 48.2 | 56.0 | 96.6 |
| LLaVA-OV w/ Dart | 50% | 0.48× | 53.2 | 40.0 | 0.36 | 78.0 | 49.0 | 56.2 | 96.9 |
| **LLaVA-OV w/ Ours** | 50% | **0.46×** | **53.4** | **40.5** | **0.43** | **78.6** | **49.7** | **56.5** | **100.4** |
| LLaVA-OV w/ FastV | 25% | 0.25× | 39.6 | 23.6 | 0.30 | 64.2 | 33.6 | 44.5 | 72.0 |
| LLaVA-OV w/ PDrop | 25%* | 0.24× | 50.8 | 38.2 | 0.35 | 76.3 | 48.2 | 53.5 | 93.6 |
| LLaVA-OV w/ Dart | 25% | 0.25× | **51.5** | 38.7 | 0.33 | 76.6 | 47.0 | **55.1** | 93.3 |
| **LLaVA-OV w/ Ours** | 25% | **0.22×** | 51.3 | **39.0** | **0.42** | **77.0** | **49.0** | 54.5 | **97.3** |
| Qwen2-VL | 100% | 1.00× | 50.4 | 51.0 | 1.23 | 76.8 | 45.5 | 53.6 | 100 |
| Qwen2-VL w/ FastV | 50% | 0.48× | 32.4 | 36.3 | 0.52 | 43.9 | 28.3 | 38.2 | 61.4 |
| Qwen2-VL w/ PDrop | 50%* | 0.47× | 48.9 | 49.6 | 1.14 | 75.2 | 45.4 | 50.8 | 96.6 |
| Qwen2-VL w/ Dart | 50% | 0.48× | 49.6 | 49.4 | 1.17 | 76.4 | 44.8 | 52.0 | 97.6 |
| **Qwen2-VL w/ Ours** | 50% | **0.46×** | **50.0** | **49.8** | **1.18** | 75.6 | **45.9** | **52.4** | **98.3** |
| Qwen2-VL w/ FastV | 25% | 0.25× | 31.2 | 36.1 | 0.48 | 42.0 | 26.8 | 35.1 | 58.5 |
| Qwen2-VL w/ PDrop | 25%* | 0.24× | 47.3 | 46.2 | 1.11 | 73.9 | 44.1 | 47.8 | 92.8 |
| Qwen2-VL w/ Dart | 25% | 0.25× | 47.4 | 47.1 | 1.10 | 74.1 | **44.5** | 49.2 | 93.7 |
| **Qwen2-VL w/ Ours** | 25% | **0.22×** | **48.1** | 46.7 | **1.11** | **74.2** | 44.3 | **50.5** | **94.2** |
| VILA | 100% | 1.00× | 47.3 | 34.0 | 1.29 | 69.9 | 46.2 | 55.6 | 100 |
| VILA w/ FastV | 50% | 0.49× | 42.2 | 20.7 | 0.98 | 62.9 | 36.8 | 47.1 | 80.1 |
| VILA w/ PDrop | 50%* | 0.49× | 47.3 | 35.0 | 1.22 | 69.4 | 45.8 | 55.1 | 99.2 |
| VILA w/ Dart | 50% | 0.49× | 46.1 | 34.7 | 1.25 | 69.2 | 46.5 | 55.2 | 99.2 |
| **VILA w/ Ours** | 50% | **0.47×** | **47.6** | **35.2** | **1.25** | **70.3** | 46.4 | **55.4** | **100.3** |
| VILA w/ FastV | 25% | 0.26× | 41.4 | 20.5 | 0.97 | 61.5 | 36.4 | 46.8 | 79.0 |
| VILA w/ PDrop | 25%* | 0.26× | 45.2 | 33.6 | 1.24 | 68.2 | 45.1 | 54.8 | 97.4 |
| VILA w/ Dart | 25% | 0.26× | 45.3 | 34.6 | 1.23 | 68.1 | 45.6 | 54.2 | 97.7 |
| **VILA w/ Ours** | 25% | **0.23×** | **45.5** | **35.6** | 1.22 | **69.4** | 46.4 | **54.8** | **99.0** |

## 4.3 EFFECTIVENESS OF EXPLAINABILITY-BASED COMPRESSOR LEARNING

The performance of the $\tilde{R}_v$-guided token compressor is evaluated in this section. $\tilde{R}_v$ is generated by the learned $f_\theta$, and the token pruning is performed accordingly before the LLM computation. Five image and six video benchmarks are included for evaluation.

**Performance Comparison.** Table 2 presents the results of LLaVA-OneVision and Qwen2-VL under different token compression retention ratios on image benchmarks. We exclude VILA here because it uses a fixed and relatively small number of image tokens, making compression less meaningful. As shown in the table, at a retention rate of 50%, our compressor demonstrates overall superiority over the baselines at comparable FLOPs, achieving average improvements of 2.6% and 1.2% across all benchmarks for LLaVA-OneVision and Qwen2-VL, respectively. When the retention rate is further reduced to 25%, the performance gains increase to 4.7% and 1.6%, highlighting the enhanced robustness of our method under higher compression rates. We conducted additional experiments on more image benchmarks; please refer to the Appendix F.

In Table 3, we evaluate the compression performance of LLaVA-OneVision, Qwen2-VL, and VILA on video benchmarks. We make several observations. First, our compressor consistently outperforms baselines at comparable FLOPs, regardless of the model and retention ratio. Both LLaVA-OneVision and VILA are able to maintain 100% performance when 50% of the visual tokens are pruned. Second, VILA exhibits the smallest performance drop, while Qwen2-VL shows the largest, likely due to its attention patterns being harder to capture. Importantly, our task-related compression can still further reduce token redundancy on both Qwen2-VL and VILA, demonstrating that it complements the task-agnostic compression already presented in these models. Finally, comparing the results in Tables 1, 2, and 3, the performance degradation from the $R_v$-guided compressor to the $\tilde{R}_v$-guided compressor is more pronounced in image tasks. This is likely also due to the greater redundancy in videos, which reduces the learning difficulty.

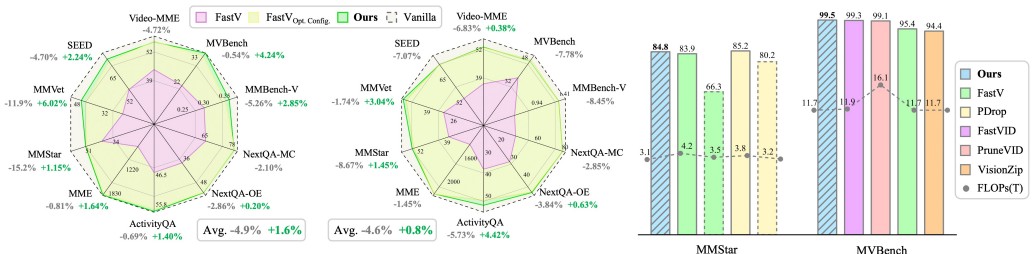

(a) Radar charts of LLaVA-OneVision(left) and Qwen2-VL(right) with 25% visual tokens retained

(b) Performance preservation(%) of methods based on LLaVA-OneVision

Figure 3: **Comparison results on larger images and longer videos.** Performance preservation ratio measures the performance retained relative to Vanilla. Gray text denotes the gap from Vanilla and green text highlights improvements over the FastV_Opt.Config..

Table 4: **Efficiency analysis based on Qwen2-VL on MMStar.** We evaluate the inference costs in terms of total inference time, prefilling time, FLOPs, and KV cache memory. KV cache memory is computed with consideration of the Grouped Query Attention (GQA) used in practical inference.

| Method | Retention Ratio | FLOPs($\times$) | Total Inference Time | Prefilling Time | KV Cache | Total Speedup | Prefilling Speedup | MMStar |
|---|---|---|---|---|---|---|---|---|
| Qwen2-VL | 100% | 1.00$\times$ | 15min24s | 6min36s | 71.2MB | 1.00$\times$ | 1.00$\times$ | 61.1 |
| Qwen2-VL w/ FastV | 25% | 0.27$\times$ | 12min19s | 4min14s | 19.7MB | 1.25$\times$ | 1.56$\times$ | 39.6 |
| Qwen2-VL w/ PDrop | 25%* | 0.25$\times$ | 12min15s | 4min10s | 18.1MB | 1.26$\times$ | 1.58$\times$ | 53.1 |
| Qwen2-VL w/ Dart | 25% | 0.30$\times$ | 12min20s | 4min16s | 21.6MB | 1.25$\times$ | 1.55$\times$ | 54.3 |
| **Qwen2-VL w/ Ours** | 25% | **0.24$\times$** | **12min16s** | **4min08s** | **17.8MB** | **1.26$\times$** | **1.60$\times$** | **55.8** |

**Applying to Larger Images and Longer Videos.** We evaluate the generalization of directly applying the trained $f_\theta$ on larger images and longer videos; experimental results are shown in Figure 3, with configurations detailed in Appendix C. Sub-figure(a) show the compression performance on 4 image and 6 video benchmarks based on LLaVA-OneVision and Qwen2-VL. Our method consistently outperforms FastV and achieves higher average performance than its optimal configuration, demonstrating strong generalization to larger images and longer videos than it seen during training. Detailed comparison results on these 10 benchmarks are provided in the Appendix F.

Sub-figure(b) show the comparisons on two challenging benchmarks, *i.e.*, MMStar and MVBench. Several strong methods are introduced for comparison: PruneVID Huang et al. (2024), FastVID Shen et al. (2025a), VisionZip Yang et al. (2024), and PyramidDrop Xing et al. (2024). Our approach achieves SOTA performance with the lowest FLOPs. Remarkably, even directly applying the trained $f_\theta$ to longer videos with more frames, it still performs favorably compared to methods specifically designed for videos (i.e., FastVID and PruneVID). Beyond superior performance, our lightweight compressor significantly improves MLLM inference efficiency with negligible additional cost. We follow Dart Wen et al. (2025) and report efficiency in terms of total inference time, prefilling time, FLOPs, and KV cache memory, as shown in Table 4. Our compressor achieves both the best performance and highest efficiency. The prefill-stage acceleration (1.60$\times$) and reduced KV cache footprint (71.2MB $\rightarrow$17.8MB) enable efficient processing in prefill and decode stages, keeping total inference time comparable to other methods (see Appendix F for more results).

**Analysis and Discussion.** We find it interesting that a lightweight $f_\theta$ can achieve strong performance, thus we offer further discussion here: (i) *The convolutional mapper $f_\theta$ learns a structurally matched input–output mapping.* As shown in Eq. 1, the relevance map $R_t$ is essentially an iterative composition of attention maps $A_t^l$ with their task-specific gradients across layers. Therefore, using $A$ to predict $R$ is natural and intuitively well-motivated. Visualization results in Appendix B show that $f_\theta$ exhibits task-sensitive behavior, adaptively focusing on the relevant visual regions according to the instruction. (ii) *Shallow-layer attention maps contain vital information.* It has been demonstrated in several works Chen et al. (2024b); Xing et al. (2024) that visual tokens make a greater contribution to output generation in the shallow layers compared to the deeper layers. Our experimental results are consistent with the finding. First, using the first-layer attention map as the input of $f_\theta$ already provides a strong guidance for token compression. Moreover, the attention maps from shallow layers perform well overall; for detailed ablation results, please refer to the Appendix E. (iii) *We aim not to learn an identical $R_t$, but rather its relatively large values.* The loss function design in Section 3.3 masked the bottom 50% of label values, simplifying the learning task to the

Table 5: **Ablation study on explainability methods for relevance map generation.** We evaluate two strategies for aggregating multi-head attention maps—gradient-weighted summation and simple averaging—to generate relevance maps for token compression on video and image benchmarks.

| Model | Method | Rentation Ratio | Image Benchmark | | | Video Benchmark | | | Avg.(%) |
|---|---|---|---|---|---|---|---|---|---|
| | | | MME | MMStar | MMVet | Video-MME | MVBench | MMBench-V | |
| LLaVA-OneVision | Vanilla | 100% | 1997.7 | 60.5 | 48.7 | 53.6 | 41.2 | 0.41 | 100 |
| | Mean-weighted | 50% | 1974.5 | 58.5 | 45.9 | 53.6 | 40.8 | 0.39 | 97.3 |
| | **Grad-weighted** | | 1974.2 | 59.7 | 47.2 | 54.3 | 41.1 | 0.40 | **98.8** |
| Qwen2-VL | Vanilla | 100% | 2295.1 | 60.4 | 54.0 | 50.4 | 51.0 | 1.23 | 100 |
| | Mean-weighted | 50% | 2300.6 | 58.2 | 49.2 | 49.9 | 49.9 | 1.15 | 96.3 |
| | **Grad-weighted** | | 2297.1 | 60.3 | 53.2 | 51.0 | 50.7 | 1.19 | **99.3** |
| VILA1.5 | Vanilla | 100% | 1700.3 | 38.7 | 39.3 | 47.3 | 34.0 | 1.29 | 100 |
| | Mean-weighted | 50% | 1720.8 | 38.0 | 34.2 | 48.0 | 34.1 | 1.20 | 96.9 |
| | **Grad-weighted** | | 1740.5 | 37.2 | 38.0 | 47.9 | 34.2 | 1.26 | **99.1** |

distribution of the top 50%. Our goal is not to learn a precise one-to-one input–output mapping. Instead, the learning target is only to achieve identifying which regions are relevant to the instruction, thereby enabling token pruning. One can observe the visualization results in Appendix B for evidence. The pruning results $\hat{V}$ (based on $R_v$) and the convolutional network's learned $\hat{\tilde{V}}$ (based on $\tilde{R}_v$) are not identical, yet it retains the same task-related regions and produces correct answers.

## 4.4 ABLATION STUDY

As shown in Eq. 1, the relevance map is updated by aggregating attention maps across layers, where the multiple heads in each layer are combined either via simple averaging or gradient-weighted averaging (used in our approach). Table 5 shows that employing gradient-weighted aggregation to generate $R_v$ for token compression performs consistently better than simple averaging across image and video benchmarks. A reasonable explanation is that differing contributions of attention heads make simple averaging prone to distorting relevance maps Voita et al. (2019). We also include ablation studies of different configurations for training $f_\theta$ in Appendix E, specifically investigating (i) the effect of varying the depth of the convolutional network and (ii) the influence of selecting different attention layer as input for the compressor.

## 5 CONCLUSION

In this work, we demonstrate the feasibility of task-related visual token compression at the LLM input stage. We first demonstrate experimentally that the relevance scores derived from explainability methods well evaluate the task-related importance of visual tokens, which can be used for effective token compression. To enable efficient and practical deployment, we employ a simple convolutional network to learn a mapping from the LLM first-layer attention maps to the explainability-derived relevance scores. Using the predicted relevance scores from lightweight model, token compression can be performed prior to the LLM. Extensive experiments demonstrate the effectiveness and generalizability of our task-related token compression method. Since the relevance scores are obtained via backward computations, their generation is resource-intensive. This poses a challenge in scaling the compressor training to high-resolution images or long video sequences. Future work will explore stronger compressors and the use of relevance scores to guide token compression during training.

**Reproducibility Statement.** We provide the necessary information to facilitate reproducibility. Experimental settings and implementation details are described in the Section 4.1 and Appendix C. All the datasets used in our experiments are publicly available, and the preprocessing steps of training data are documented in Appendix C.

## ACKNOWLEDGEMENTS

This work was supported in part by the grants from National Science and Technology Major Project (No. 2023ZD0121104), and National Natural Science Foundation of China (No.62222213, U22B2059)

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

APPENDIX

## A  DETAILS OF RELEVANCE PROPAGATION EQUATION

The relevance propagation in Eq. 1 follows the Generic Attention Explainability (GAE) framework Chefer et al. (2021b), which is a powerful method to interpret predictions for Transformer-based architectures. We trace the contribution of each visual token to the model's output by leveraging the self-attention modules within the GAE framework to assess token importance. Specifically, GAE first generates a relevance map $\bar{A}^l$ for each layer $l$ by integrating raw attention map and its gradients with respect to the current output $y$:

$$\bar{A}^l = \mathbb{E}_h((A^l \odot \nabla A^l)^+), \tag{4}$$

where $A^l$ can be obtained through a forward pass and the related gradient $\nabla A^l := \frac{\partial y_t}{\partial A^l}$ can be cached during a backward pass. GAE computes the Hadamard product of the attention map and its gradient for the intuitive reasons that: 1) it captures how much attention each token receives from other tokens (information from the attention map), 2) it identifies which tokens require more attention to effectively influence the output (information from the gradient). $()^+$ represents the operation of setting negative values to 0. By zeroing out negative gradient components, GAE enforces a causal direction of influence and prevents the suppression of informative positive signals by accumulated negative components Chefer et al. (2021a); Barkan et al. (2021). Gradient can also be viewed as weight in the head aggregation to indicate corresponding importance Voita et al. (2019), thereby $\mathbb{E}_h$ performs a gradient-weighted expectation over the head dimension.

Since the attention module is followed by a residual connection, GAE accumulate the relevance map by adding each layer's contribution $\bar{A}^l$ to the aggregated relevance map $R$:

$$R = R + \bar{A}^l \cdot R, \tag{5}$$

The overall relevance map $R$ is initialized as the identity matrix with the intuition that each input token's relevance score is identical in the beginning. Then the relevance propagation updates the $R$ from the 0-th layer to the last layer in the Transformer.

The traditional GAE method lacks a mechanism for handling the token-by-token autoregressive outputs required by MLLMs. Therefore, we adopt the stepwise relevance computation from R-GAE Yao et al. (2024), a GAE-derived method adapted to MLLMs. For a generated sequence of textual tokens $Y = \{y_0, y_1, \ldots, y_{T-1}\}$, we obtain the overall relevance by averaging the GAE relevance maps computed using Eq. 5 at each decoding step $t$.

## B  MORE VISUALIZATION RESULTS

### B.1  VISUALIZATION RESULTS ACROSS DIFFERENT MLLMS

We present visualization results for LLaVA-OneVision, Qwen2-VL, and VILA1.5 on both video and image inputs in Figures 4-8. Given an input image or video $V$, we first show the visual relevance scores $R_v$ with respect to the current response obtained using an explainability method. Based on $R_v$, we visualize the results of token pruning at 50% and 25% retention ratios (labeled as Top-50% compressed $\hat{V}$ and Top-25% compressed $\hat{V}$ in the figures). Then, we visualize the pruning results produced by our trained compressor ($f_\theta$) under the same compression ratios (labeled as Top-50% compressed $\hat{v}$ and Top-25% compressed $\hat{v}$ in the figures).

### B.2  CASE STUDY: EXPLAINABILITY REVEALS INSTRUCTION-RELATED VISUAL TOKENS

To demonstrate the effectiveness of explainability methods in identifying visual tokens that are highly relevant to instructions, we present 2 case studies covering both video and image inputs.

Given the same input $V$, the explainability method generates visual relevance scores $R_v$ that selectively emphasize different visual tokens according to varying user instructions. As shown in Figure 9, when the user instruction specifically targets clothing-related information, the visual tokens corresponding to the person's clothing in the video obtain higher relevance scores compared to

*User Instruction: According to the video, what do the three curved lines extending from bottom up symbolize? A. Heat flow, B. Stream., C. Vapor., D. Air. Response: C. Vapor*

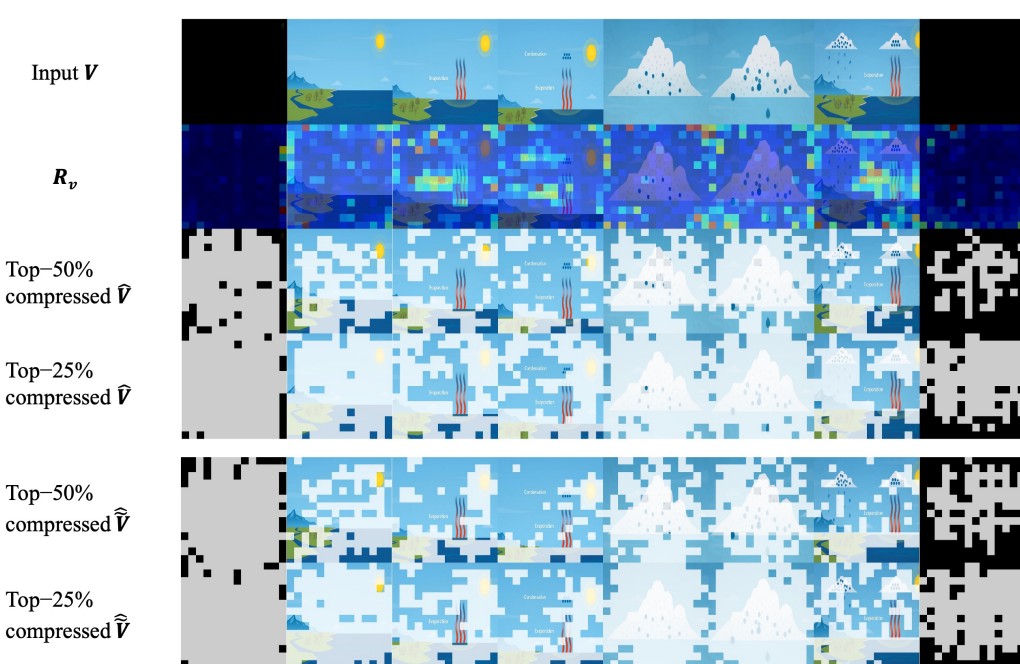

Figure 4: **Video Input Visualizations for LLaVA-OneVision**.

*User Instruction: What is the highest fueling cost?*
*Response: The highest fueling cost, as indicated by the bar chart, is for the Ford F150, which is $130.96.*

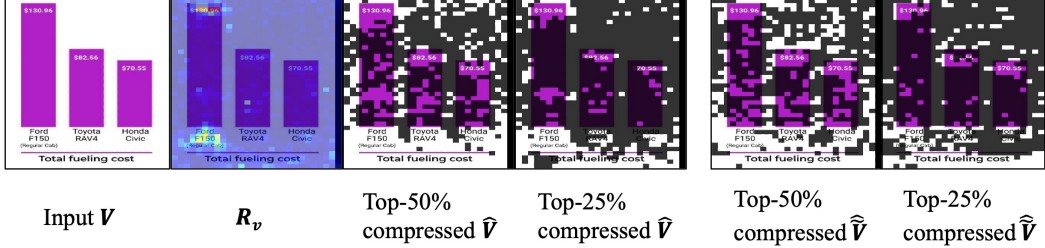

Figure 5: **Image Input Visualizations for LLaVA-OneVision**.

instructions requesting a general summary. Similarly, in Figure 10, visual tokens relevant to the user instruction exhibit higher relevance scores. When the user instruction specifies excluding the Ford F150, the visual attention shifts primarily to the other two columns. In contrast, when the instruction highlights the highest fueling cost, the Ford F150 column attracts nearly all the attention.

From a visualization standpoint, we further corroborate that the explanation results faithfully reflect the critical visual information required by the MLLM to answer the question.

## C    IMPLEMANTATION DETAILS.

**Generating** $R_v$. To derive $R_v$, our implementation employs eager attention, allowing access to full-layer attention maps required by the explainability method Chefer et al. (2021b). Compared to FlashAttention Dao et al. (2022) and inference based on KV cache Pope et al. (2023), eager attention requires more memory. To avoid out-of-memory errors and ensure efficient data generation, we limit the number of visual tokens to approximately 1500 per sample. Specifically, for video inputs, LLaVA-OneVision, VILA and Qwen2-VL are all set to sample 8 frames, resulting in 1569, 1568

*User Instruction*: *According to the video, what do the three curved lines extending from bottom up symbolize? A. Heat flow., B. Stream., C. Vapor., D. Air.* **Response**: *C. Vapor*

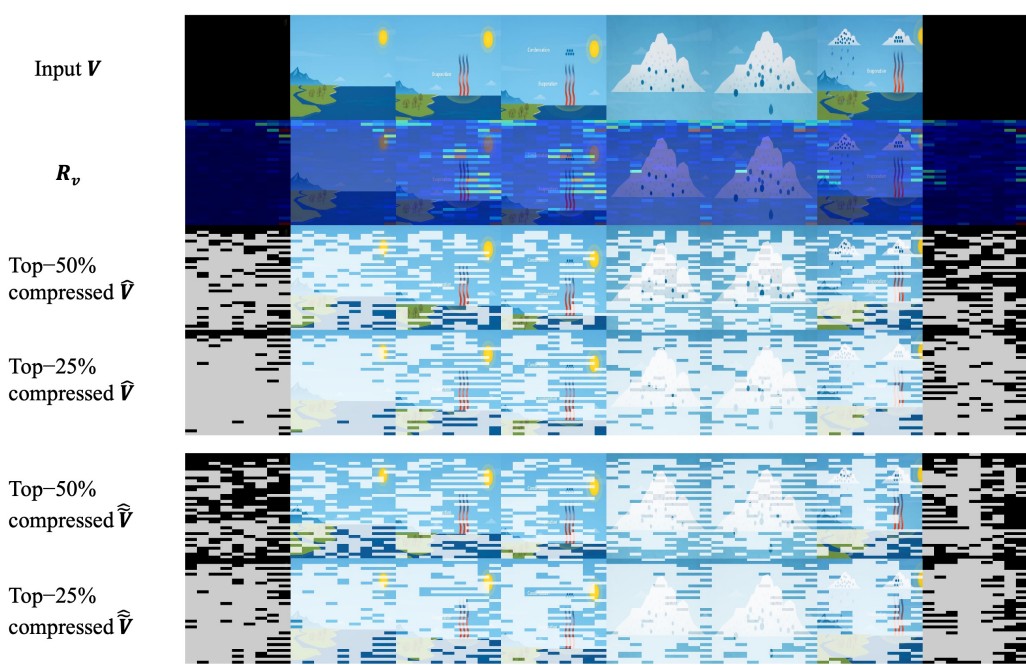

Figure 6: **Video Input Visualizations for Qwen2-VL**.

*User Instruction*: *What is the highest fueling cost?*
*Response*: *The highest fueling cost, as indicated by the bar chart, is for the Ford F150, which is $130.96.*

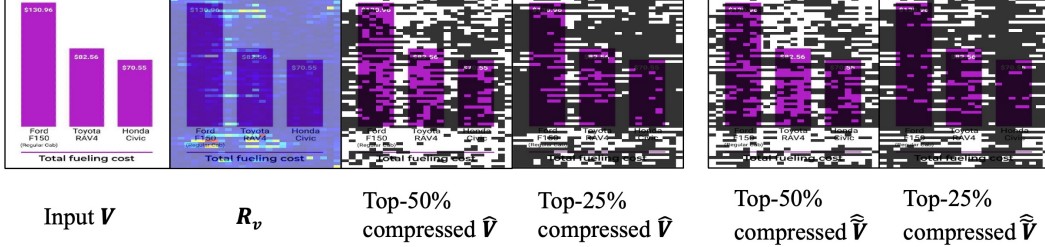

Figure 7: **Image Input Visualizations for Qwen2-VL**.

and 1296 visual tokens per video, respectively. For image inputs, LLaVA-OneVision and Qwen2-VL use similar image resolutions, resulting in 1849 and 1500 visual tokens per image, respectively. VILA always processes an image as 196 tokens, eliminating the need for additional configuration. The generated $R_v$ can be used directly to guide token pruning or to train $f_\theta$.

**Training $f_\theta$.** $f_\theta$ is implemented as a five-layer fully convolutional network with channel dimensions of $32, 64, 128, 256$, and $512$. Each layer employs a 1D depthwise separable convolution Chollet (2017), *i.e.*, a depthwise convolution with a kernel size of 3 followed by a pointwise convolution. An additional pointwise convolution layer is applied at the end for channel aggregation. The network is trained by using Adam Kingma & Ba (2015) with default settings and a batch size of 128. Training data is collected from open-source datasets: a subset of LLaVA-Video Zhang et al. (2024c) for videos and a subset of Infinity-MM Gu et al. (2024) for images, each containing approximately 10K samples. Note that $f_\theta$ is specific to MLLM, so each MLLM generates its own $A_v^0$ and $R_v$ based on the input image- or video-text pair for training. The training is performed for roughly 100 epochs, taking about half an hour for image data and less than four hours for video data on a single A100 GPU.

*User Instruction*: According to the video, what do the three curved lines extending from bottom up symbolize? A. Heat flow., B. Stream., C. Vapor., D. Air. **Response**: C. Vapor

Input $V$

$R_v$

Top-50% compressed $\hat{V}$

Top-25% compressed $\hat{V}$

Top-50% compressed $\hat{\hat{V}}$

Top-25% compressed $\hat{\hat{V}}$

Figure 8: **Video Input Visualizations for VILA**.

Input $V$

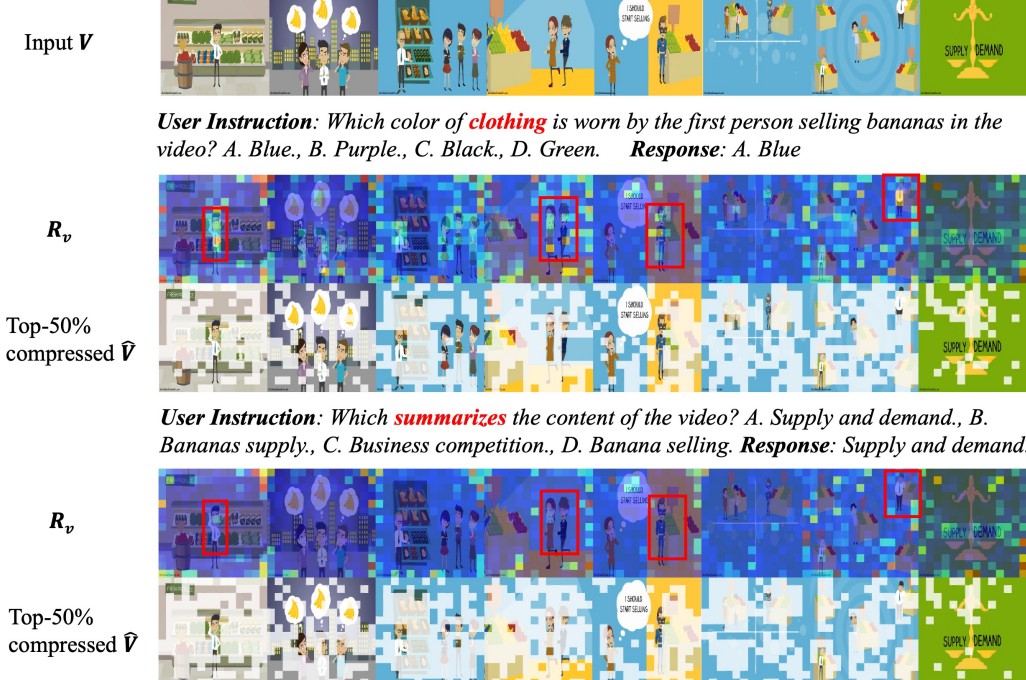

*User Instruction*: Which color of **clothing** is worn by the first person selling bananas in the video? A. Blue., B. Purple., C. Black., D. Green. **Response**: A. Blue

$R_v$

Top-50% compressed $\hat{V}$

*User Instruction*: Which **summarizes** the content of the video? A. Supply and demand., B. Bananas supply., C. Business competition., D. Banana selling. **Response**: Supply and demand.

$R_v$

Top-50% compressed $\hat{V}$

Figure 9: **Case Study 1**.

**Details of Data for Training** $f_\theta$**.** We train our explainability-based compressor based on subsets sampled from high-quality open-source datasets. First, the details of the sampling are as follows:

*User Instruction*: *What is the average total fueling cost **excluding the Ford F150**? **Response**: 76.55*

*User Instruction*: *What is the **highest** fueling cost? **Response**: 130.96*

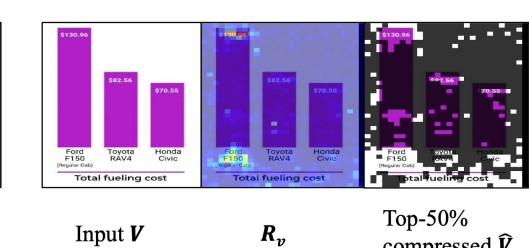

Figure 10: **Case Study 2**.

*Image Dataset.* For training the compressor used in image tasks, we sample a subset of Infinity-MM that ensures high quality and diversity. The training set primarily consists of data used during Stage 4, including 9k samples randomly sampled from the *Data Generated by GPT-4* subset and 4k from *Synthetic Data*.

*Video Dataset.* For training the compressor used in video tasks, we sample a subset of LLaVA-Video. Specifically, we include 7k samples from *LLaVA-Video*, 6k from *NeXT-QA* and 4k from *ActivityNetQA*. Note that the training sets of *NeXT-QA* and *ActivityNetQA* have no overlap with the testing sets used in the evaluation. During sampling, since LLaVA-Video contains several parts categorized by task type (open-ended and multi-choice) and video duration (0–30s, 30-60s, 1–2min and 2-3min), we ensure a balanced distribution by randomly selecting an equal number of training examples from each part.

Moreover, we assume that the visual attention distributions ($R_v$) associated with correct answers exhibit higher quality than those that lead to incorrect answers. Therefore, when training $f_\theta$ for a specific MLLM, the sampled data are evaluated by this MLLM, and the samples with incorrect answers are filtered out. Only samples for which the MLLM produces correct answers are retained and used as training data. The number of the retained samples ranges from 8K to 12K.

**Inference**. The learned $f_\theta$ can be seamlessly integrated into existing inference pipelines (no modifications are required for the prefill and decode phases of LLM inference). More interestingly, $f_\theta$ is capable of processing longer $A_v^0$ thanks to the fully convolution design. That is, our compression method can handle larger images and longer videos, even though the visual token number is limited to approximately 1500 during training. Corresponding experiments have been conducted. In these experiments, Qwen2-VL dynamically processes both images (with 'max_pixels' set to half of its default value) and videos (with 'VIDEO_MAX_PIXELS' and 'FPS_MAX_FRAMES' set to 384×28×28 and 32, respectively). These configurations are set to accommodate hardware resource constraints. LLaVA-OneVision also processes images dynamically with default settings, while sampling 32 frames per video as in Huang et al. (2024) for a fair comparison. For VILA, the input image size cannot be changed, and the number of input video frames is set to 16.

## D    EFFICIENCY ANALYSIS IN INFERENCE

To evaluate computational efficiency during inference, we report the FLOPs of the visual token part. Specifically, we consider the FLOPs of the multihead attention and the feed-forward network (FFN) modules as:

$$\text{FLOPs}_{\text{layer}} = 4nd^2 + 2n^2d + lnm, \tag{6}$$

where $n$ is the number of visual tokens, $d$ is the hidden state size, $m$ is the intermediate size of the FFN, and $l$ is the number of layers in the FFN. To compute the total FLOPs for the entire LLM, we simply multiply Eq. 6 by the number of Transformer layers $N_L$, i.e., $\text{FLOPs}_{\text{LLM}} = N_L(4nd^2 + 2n^2d + lnm)$.

At the input stage of the LLM, our compressor introduces additional computation. First, we consider the FLOPs introduced by the first-layer attention map:

$$\text{FLOPs}_{\text{attn}} = nd^2 + nd. \tag{7}$$

Note that only the key projection computation for visual tokens and the attention computation from textual tokens to visual tokens are required, corresponding to the term $nd^2$ and $nd$, respectively. Only the FLOPs incurred by the visual part are included.

Next, we account for the FLOPs introduced by the 1D depthwise separable convolution:

$$\text{FLOPs}_{\text{conv}} = \sum_{l=1}^{L} n(C_{in}^l k + C_{in}^l C_{out}^i), \tag{8}$$

where $C_{in}^l$ and $C_{out}^l$ denote the number of input and output channels of the l-th layer, respectively. We ensure that the output shape of each convolutional layer remains the same as its input by applying appropriate padding with respect to the kernel size $k$. As a result, the number of visual tokens $n$ remains constant across all layers. Then the total FLOPs is computed as the sum of the operations across all $L$ convolutional layers.

To intuitively understand the additional computational cost introduced by our method, we adopt a typical parameter configuration used in MLLMs. Specifically, we set the number of visual tokens $n$ to 1568, the hidden dimension $d$ to 3584, the intermediate size $m$ to 18944, and assume 3 layers per FFN block ($l = 3$). For the full LLM, we consider a 28-layer Transformer blocks ($N_L = 28$). For $f_\theta$, we follow the configuration described in Section 4.1 (Experimental Setup). Concretely, the convolutional network consists of 5 layers ($L = 5$) with kernel size $k = 3$, and channel dimensions increasing across layers: 32, 64, 128, 256, and 512. Based on these settings, $\text{FLOPs}_{\text{attn}}$ amounts to approximately 0.02 trillion, $\text{FLOPs}_{\text{conv}}$ is approximately 0.0003 trillion, while $\text{FLOPs}_{\text{LLM}}$ reaches approximately 11.69 trillion. It can be observed that the computational overhead introduced by our compressor is negligible. The computational costs of these two parts account for only **0.17%** and **0.0026%** of the total computational cost, respectively.

The FLOPs reported in the Table 2, Table 3, Table 4 and Figure 3 are computed using a standardized input setting. For image input, FLOPs are computed using a $384 \times 512$ input image as the reference (the number of visual tokens $n$ is 1728 for LLaVA-OneVision and 1302 for Qwen2-VL). For video input, LLaVA-OneVision and VILA1.5 sample 32 and 16 frames, respectively, resulting in visual token counts $n$ of 6272 and 3136. We fix Qwen2-VL's input to 32 frames at $720 \times 1280$ resolution ($n$=5824) for FLOPs calculation.

## E    ADDITIONAL ABLATION RESULTS

We perform additional ablation studies to validate the design choices of our method, specifically examining (i) the depth of the convolutional network and (ii) the attention layer index used as input. For the convolutional depth, we conduct controlled experiments on Qwen2-VL using networks of 3, 5, 7, and 10 layers, while keeping all other factors fixed, including the training data, input representation $A^0$, and training configurations (learning rate, batch size, and number of epochs). All variants follow a consistent channel-growth strategy in which the network starts with 32 channels and gradually increases capacity, with intermediate channel dimensions repeated to moderate growth and avoid over-parameterization. The exact configurations are 3-layer: [32, 64, 128], 5-layer: [32, 64, 128, 256, 512], 7-layer: [32, 64, 128, 128, 256, 256, 512], and 10-layer: [32, 64, 128, 128, 256, 256, 512, 512, 512, 512]. For the study on attention-layer index, we train our lightweight model on Qwen2-VL using attention maps extracted from different transformer layers as input, specifically evaluating layers 1, 2, 4, and 6.

The results are reported in Table 6 and Table 7, respectively. Table 6 shows that the 5-layer model achieves the best performance under retention ratios of 50% and 25%. Increasing the depth to 7 or 10 layers leads to a slight performance degradation—especially under the more challenging 25% retention condition. This suggests that deeper networks begin to overfit or suffer from optimization difficulties given the simplicity of the task. In contrast, the 3-layer model lacks sufficient capacity to capture the relevance distribution. Therefore, the 5-layer architecture strikes the optimal balance between model capacity and task complexity, which justifies our design choice.

Table 7 shows that using multi-layer attention scores as input yields only marginal gains—+0.1% average at 50% retention (with 2 layers) and +0.6% at 25% retention (with 4 layers)—yet incurs substantial computational overhead, with FLOPs increasing from 0.24× to 0.43× at 25% retention, almost two times. Notably, the first-layer input achieves nearly the same performance as multi-layer

Table 6: **Ablation study on the depth of the convolutional network .** In our work, we use a 5-layer convolutional network as the explainability-based compressor before the LLM.

| Methods | Conv Depth | Retention Ratio | FLOPs | Image Benchmark | | | Video Benchmark | | | Avg.(%) |
|---|---|---|---|---|---|---|---|---|---|---|
| | | | | MME | MMStar | MMVet | Video-MME | MVBench | MMBench-V | |
| Qwen2-VL | - | 100% | 1.00× | 2295.1 | 60.4 | 54.0 | 50.4 | 51.0 | 1.23 | 100 |
| Qwen2-VL w/ Ours | 3 | 50% | 0.49× | 2265.0 | 55.5 | 50.4 | 49.8 | 49.3 | 1.15 | 95.5 |
| | **5** | 50% | 0.49× | 2288.3 | 55.9 | 51.9 | 50.0 | 49.8 | 1.18 | **96.9** |
| | 7 | 50% | 0.49× | 2279.4 | 56.1 | 51.9 | 49.8 | 49.4 | 1.17 | 96.5 |
| | 10 | 50% | 0.49× | 2275.1 | 55.1 | 51.5 | 50.0 | 49.7 | 1.15 | 96.0 |
| | 3 | 25% | 0.24× | 2297.1 | 60.3 | 53.2 | 51.0 | 50.7 | 1.19 | 90.7 |
| | **5** | 25% | 0.24× | 2299.1 | 58.7 | 51.7 | 50.3 | 49.7 | 1.17 | **91.7** |
| | 7 | 25% | 0.24× | 2299.1 | 58.7 | 51.7 | 50.3 | 49.7 | 1.17 | 91.1 |
| | 10 | 25% | 0.24× | 2299.1 | 58.7 | 51.7 | 50.3 | 49.7 | 1.17 | 90.4 |

Table 7: **Ablation study on the attention layer index used as input.** In our work, the explainability-based compressor takes the first-layer attention map $A^0$ as its input.

| Methods | Input Layer | Retention Ratio | FLOPs | Image Benchmark | | | Video Benchmark | | | Avg.(%) |
|---|---|---|---|---|---|---|---|---|---|---|
| | | | | MME | MMStar | MMVet | Video-MME | MVBench | MMBench-V | |
| Qwen2-VL | - | 100% | 1.00× | 2295.1 | 60.4 | 54.0 | 50.4 | 51.0 | 1.23 | 100 |
| Qwen2-VL w/ Ours | **1** | 50% | **0.49×** | 2288.3 | 55.9 | 51.9 | 50.0 | 49.8 | 1.18 | 96.9 |
| | 2 | 50% | 0.53× | 2279.8 | 55.8 | 52.0 | 50.1 | 50.2 | 1.18 | **97.0** |
| | 4 | 50% | 0.60× | 2303.0 | 56.3 | 52.2 | 49.3 | 50.1 | 1.15 | 96.6 |
| | 6 | 50% | 0.67× | 2278.1 | 56.7 | 49.5 | 49.4 | 49.6 | 1.19 | 96.1 |
| | **1** | 25% | **0.24×** | 2280.9 | 51.8 | 47.3 | 48.1 | 46.7 | 1.11 | 91.7 |
| | 2 | 25% | 0.35× | 2272.0 | 51.9 | 46.6 | 47.8 | 48.4 | 1.10 | 91.7 |
| | 4 | 25% | 0.35× | 2270.2 | 52.1 | 48.0 | 48.0 | 48.1 | 1.11 | **92.3** |
| | 6 | 25% | 0.43× | 2252.3 | 52.1 | 47.2 | 47.4 | 47.6 | 1.11 | 91.6 |

variants while being significantly more efficient. Our work aims to achieve an innovative paradigm shift that enables task-related token compression to be applied prior to the LLM, which not only significantly reduces computation and memory overhead during both prefill and decode phases, but also allows deployment without modifying the LLM architecture. However, leveraging deeper-layer attention maps as input yields only marginal improvements while contradicting our design goal. Overall, the first-layer attention map suffices for our purpose.

## F ADDITIONAL EXPERIMENTAL RESULTS

For image tasks, we further conducted evaluations across additional benchmarks, including GQA Hudson & Manning (2019) (real-world visual reasoning), ScienceQA (SQA) Lu et al. (2022) (scientific reasoning) , VizWiz Gurari et al. (2018) (real-world robustness), ChartQA Masry et al. (2022) (chart reasoning), DocVQA Mathew et al. (2021) (document reasoning), OCRBench Liu et al. (2024b) (OCR reasoning), to comprehensively assess the effectiveness of our method. The results are presented in Table 8, our method consistently outperforms the baselines across diverse image benchmarks, including real-world visual reasoning and OCR-related tasks.

For video tasks, we also investigate a more aggressive compression setting with a 10% retention ratio in Table 9, as a supplement to Table 3. Our method attains the lowest FLOPs while preserving competitive accuracy, achieving average improvements of 3.7%, 3.2%, and 1.9% across all benchmarks for LLaVA-OneVision, Qwen2-VL, and VILA, respectively, compared to the best-performing baseline. Notably, even under such extreme compression, our method consistently delivers strong results, highlighting its robustness across different MLLMs.

We provide full tables of results corresponding to the generalization experiments shown in the Figure 3 (a) in the main text (Applying to Larger Images and Longer Videos), with detailed results for the image and video benchmarks listed in Table 10 and Table 11, respectively. In addition, we provide detailed comparison results shown in the Figure 3 (b) for two challenging benchmarks, MMStar and MVBench, in Table 12 and Table 13.

Table 8: **Compare explainability-based compressor on image benchmarks.** Values marked with * in Retention Ratio denote the average retention ratio across LLM layers due to multi-stage compression in PDrop.

| Method | Retention Ratio | FLOPs | GQA | SQA | VizWiz | ChartQA | DocVQA | OCRBench |
|---|---|---|---|---|---|---|---|---|
| LLaVA-OneVision | 100% | 1.00× | 63.1 | 95.1 | 33.1 | 71.4 | 73.3 | 54.4 |
| LLaVA-OneVision w/ FastV | 50% | 0.51× | 42.5 | 76.2 | 3.2 | 41.4 | 42.9 | 28.3 |
| LLaVA-OneVision w/ Pdrop | 51%* | 0.51× | 61.0 | 92.0 | 29.7 | 62.0 | 58.0 | 42.0 |
| LLaVA-OneVision w/ Dart | 50% | 0.51× | **61.7** | 91.7 | 29.8 | 61.0 | 55.3 | 43.2 |
| **LLaVA-OneVision w/ Ours** | 50% | **0.48×** | 61.4 | **92.2** | **33.6** | **63.0** | **62.0** | **47.0** |
| LLaVA-OneVision w/ FastV | 25% | 0.27× | 39.9 | 70.4 | 2.7 | 21.3 | 22.2 | 13.9 |
| LLaVA-OneVision w/ PDrop | 25%* | 0.25× | 57.2 | 87.9 | 25.3 | 43.4 | 34.9 | 30.2 |
| LLaVA-OneVision w/ Dart | 25% | 0.27× | 57.5 | 89.1 | 27.9 | 46.7 | 39.9 | 34.3 |
| **LLaVA-OneVision w/ Ours** | 25% | **0.24×** | **59.0** | **90.2** | **33.0** | **52.7** | **49.4** | **38.3** |
| Qwen2-VL | 100% | 1.00× | 62.2 | 85.7 | 44.3 | 92.7 | 93.1 | 81.5 |
| Qwen2-VL w/ FastV | 50% | 0.51× | 43.9 | 65.3 | 32.5 | 51.3 | 59.2 | 56.3 |
| Qwen2-VL w/ PDrop | 51%* | 0.51× | 60.8 | 83.5 | 41.9 | 77.0 | 78.5 | 78.8 |
| Qwen2-VL w/ Dart | 50% | 0.51× | 61.3 | 84.7 | 43.9 | 78.4 | 79.7 | 78.5 |
| **Qwen2-VL w/ Ours** | 50% | **0.49×** | **61.6** | **85.5** | **44.1** | **80.1** | **81.2** | **79.5** |
| Qwen2-VL w/ FastV | 25% | 0.27× | 41.5 | 64.3 | 31.5 | 41.1 | 46.1 | 46.8 |
| Qwen2-VL w/ PDrop | 25%* | 0.25× | 57.1 | 83.2 | 40.8 | 61.7 | 58.2 | 63.9 |
| Qwen2-VL w/ Dart | 25% | 0.27× | 58.8 | 83.9 | 41.3 | 65.0 | 62.8 | 65.1 |
| **Qwen2-VL w/ Ours** | 25% | **0.24×** | **59.3** | **84.3** | **43.1** | **70.9** | **72.1** | **67.8** |

Table 9: **Compare explainability-based compressor on video benchmarks.**

| Method | Retention Ratio | FLOPs | Video-MME | MVBench | MMBench-Video | Next-QA multi-choice | Next-QA open-ended | Activity-QA | Avg.(%) |
|---|---|---|---|---|---|---|---|---|---|
| LLaVA-OV | 100% | 1.00× | 53.6 | 41.2 | 0.41 | 79.2 | 49.0 | 56.9 | 100 |
| LLaVA-OV w/ FastV | 10% | 0.12× | 37.7 | 22.4 | 0.27 | 60.3 | 30.6 | 39.9 | 66.5 |
| LLaVA-OV w/ PDrop | 10%* | 0.10× | 47.0 | 37.0 | 0.35 | 72.3 | 43.7 | 50.0 | 88.5 |
| LLaVA-OV w/ Dart | 10% | 0.12× | **47.3** | 36.9 | 0.36 | 72.7 | 43.5 | 50.3 | 89.1 |
| **LLaVA-OV w/ Ours** | 10% | **0.09×** | 47.1 | **37.4** | **0.40** | **76.5** | **45.6** | **51.6** | **92.8** |
| Qwen2-VL | 100% | 1.00× | 50.4 | 51.0 | 1.23 | 76.8 | 45.5 | 53.6 | 100 |
| Qwen2-VL w/ FastV | 10% | 0.12× | 29.1 | 37.5 | 0.44 | 39.4 | 23.3 | 32.0 | 54.9 |
| Qwen2-VL w/ PDrop | 10%* | 0.10× | 45.2 | 40.3 | 0.82 | 71.5 | 41.8 | 45.1 | 84.1 |
| Qwen2-VL w/ Dart | 10% | 0.12× | 45.8 | 41.6 | 0.85 | **72.1** | 42.6 | 45.5 | 85.7 |
| **Qwen2-VL w/ Ours** | 10% | **0.09×** | **46.1** | **42.5** | **1.00** | 72.0 | **43.3** | **47.5** | **88.9** |
| VILA | 100% | 1.00× | 47.3 | 34.0 | 1.29 | 69.9 | 46.2 | 55.6 | 100 |
| VILA w/ FastV | 10% | 0.12× | 37.8 | 19.5 | 0.88 | 57.9 | 33.9 | 43.2 | 73.2 |
| VILA w/ PDrop | 10%* | 0.11× | 43.0 | 34.4 | 1.13 | 65.2 | 43.5 | 50.6 | 93.0 |
| VILA w/ Dart | 10% | 0.12× | 42.9 | 34.8 | 1.14 | 64.8 | 43.7 | 51.1 | 93.4 |
| **VILA w/ Ours** | 10% | **0.09×** | **43.6** | **35.0** | **1.14** | **67.0** | **44.7** | **53.0** | **95.3** |

Table 14 exhibits the additional efficiency analysis on MMVet. Our lightweight compressor achieves substantial reductions in KV-cache usage and accelerates the prefill stage, while achieving the highest task scores and keeping overall inference time comparable to baselines. These results demonstrate that our approach maintains both strong task performance and computational efficiency.

# G  THE USE OF LARGE LANGUAGE MODELS(LLMs)

In preparing this manuscript, we used a large language model (LLM, specifically GPT-5-mini) solely as a general-purpose writing and editing assistant. The LLM was employed to improve clarity, grammar, and overall presentation of the text. All technical content, experiments results, and interpretations were generated and verified by the authors. The LLM did not contribute to research ideation, experimental design, data analysis, or the writing of original technical content. The authors take full responsibility for all content presented in this paper.

Table 10: **Compare generalization performance of our compressor on image benmarks.**

| Method | Rentention Ratio | FLOPs | MME | MMStar | MMVet | SEED | Avg.(%) |
|---|---|---|---|---|---|---|---|
| LLaVA-OneVision | 100% | 1.00× | 2002.0 | 62.0 | 52.0 | 76.7 | 100 |
| LLaVA-OneVision w/ FastV$_{Opt.Config.}$ | 50% | 0.51× | 1990.3 | 57.3 | 48.4 | 75.7 | 95.9 |
| **LLaVA-OneVision w/ Ours** | 50% | 0.48× | 1988.0 | 57.8 | 50.2 | 75.4 | **96.8** |
| LLaVA-OneVision w/ FastV$_{Opt.Config.}$ | 25% | 0.27× | 1953.7 | 52.0 | 43.2 | 71.5 | 89.4 |
| **LLaVA-OneVision w/ Ours** | 25% | 0.24× | 1985.8 | 52.6 | 45.8 | 73.1 | **91.9** |
| Qwen2-VL | 100% | 1.00× | 2316.6 | 61.1 | 51.7 | 76.4 | 100 |
| Qwen2-VL w/ FastV$_{Opt.Config.}$ | 50% | 0.51× | 2295.8 | 57.7 | 52.4 | 74.8 | 98.2 |
| **Qwen2-VL w/ Ours** | 50% | 0.49× | 2311.7 | 57.9 | 53.9 | 73.9 | **98.9** |
| Qwen2-VL w/ FastV$_{Opt.Config.}$ | 25% | 0.27× | 2288.2 | 55.0 | 49.3 | 71.1 | 94.3 |
| **Qwen2-VL w/ Ours** | 25% | 0.24× | 2283.1 | 55.8 | 50.8 | 71.0 | **95.3** |

Table 11: **Compare generalization performance of our compressor on video benchmarks.**

| Method | Retention Ratio | FLOPs | Video-MME | MVBench | MMBench-Video | Next-QA MC | Next-QA OE | Activity-QA | Avg.(%) |
|---|---|---|---|---|---|---|---|---|---|
| LLaVA-OV | 100% | 1.00× | 59.3 | 37.1 | 0.38 | 80.9 | 52.5 | 58.4 | 100 |
| LLaVA-OV w/ FastV$_{Opt.Config.}$ | 50% | 0.48× | 58.8 | 36.1 | 0.38 | 80.5 | 51.4 | 58.2 | 98.9 |
| **LLaVA-OV w/ Ours** | 50% | 0.46× | 58.8 | 37.2 | 0.38 | 80.2 | 52.0 | 58.1 | **99.5** |
| LLaVA-OV w/ FastV$_{Opt.Config.}$ | 25% | 0.25× | 57.0 | 35.4 | 0.35 | 79.7 | 50.9 | 57.2 | 96.2 |
| **LLaVA-OV w/ Ours** | 25% | 0.22× | 56.5 | 36.9 | 0.36 | 79.2 | 51.0 | 58.0 | **97.3** |
| Qwen2-VL | 100% | 1.00× | 57.1 | 52.7 | 1.42 | 80.7 | 49.5 | 57.6 | 100 |
| Qwen2-VL w/ FastV$_{Opt.Config.}$ | 50% | 0.48× | 55.4 | 51.3 | 1.40 | 79.6 | 49.0 | 55.7 | 97.9 |
| **Qwen2-VL w/ Ours** | 50% | 0.46× | 55.7 | 51.4 | 1.41 | 79.5 | 48.7 | 56.3 | **98.2** |
| Qwen2-VL w/ FastV$_{Opt.Config.}$ | 25% | 0.25× | 53.0 | 49.6 | 1.30 | 78.6 | 47.3 | 52.0 | 93.6 |
| **Qwen2-VL w/ Ours** | 25% | 0.22× | 53.2 | 48.6 | 1.30 | 78.4 | 47.6 | 54.3 | **94.1** |
| VILA | 100% | 1.00× | 48.7 | 31.7 | 1.30 | 70.4 | 45.8 | 55.2 | 100 |
| VILA w/ FastV$_{Opt.Config.}$ | 50% | 0.49× | 48.1 | 31.5 | 1.31 | 70.1 | 46.5 | 55.1 | 100.0 |
| **VILA w/ Ours** | 50% | 0.47× | 48.4 | 34.3 | 1.34 | 70.0 | 47.0 | 56.0 | **102.4** |
| VILA w/ FastV$_{Opt.Config.}$ | 25% | 0.26× | 46.3 | 31.8 | 1.26 | 69.6 | 45.6 | 54.6 | 98.3 |
| **VILA w/ Ours** | 25% | 0.23× | 47.4 | 35.0 | 1.29 | 70.0 | 46.7 | 55.7 | **101.5** |

Table 12: **Efficiency and performance comparison across different methods on MMStar.** Values marked with * indicate that the retention ratio refers to the average proportion of retained tokens across all LLM layers, due to multi-stage compression in PDrop. For FastV, the same retention ratio corresponds to different FLOPs when compression is applied at different layers (2nd and 4th).

| Method | Retention Ratio | FLOPs(T) | Performance Preservation(%) |
|---|---|---|---|
| LLaVA-OneVision | 100% | 12.9 | 100 |
| LLaVA-OneVision w/FastV | 25.0% | 4.2 | 83.9 |
| LLaVA-OneVision w/FastV | 25.0% | 3.5 | 66.3 |
| LLaVA-OneVision w/PDrop | 30.0%* | 3.8 | **85.2** |
| LLaVA-OneVision w/PDrop | 25.4%* | 3.2 | 80.2 |
| **LLaVA-OneVision w/Ours** | 25.0% | **3.1** | 84.8 |
| Qwen2-VL | 100% | 9.6 | 100 |
| Qwen2-VL w/FastV | 25.0% | 3.1 | 90.0 |
| **Qwen2-VL w/Ours** | 25.0% | **2.4** | **91.3** |

Table 13: **Efficiency and performance comparison across different methods on MVBench.** Values marked with * indicate that the retention ratio is reported from the original paper.

| Method | Retention Ratio | FLOPs(T) | Performance Preservation(%) |
|---|---|---|---|
| LLaVA-OneVision | 100% | 52.7 | 100 |
| LLaVA-OneVision w/FastVID | 25.0% | 11.7 | 99.3 |
| LLaVA-OneVision w/PruneVID | 17.0%* | 11.9 | 99.1 |
| LLaVA-OneVision w/FastV | 25.0% | 16.1 | 95.4 |
| LLaVA-OneVision w/VisionZip | 25.0% | 11.7 | 94.4 |
| **LLaVA-OneVision w/Ours** | 25.0% | **11.7** | **99.5** |
| Qwen2-VL | 100% | 48.4 | 100 |
| Qwen2-VL w/FastV | 25.0% | 14.9 | **94.1** |
| **Qwen2-VL w/Ours** | 25.0% | **10.9** | 92.2 |
| VILA1.5 | 100% | 27.0 | 100 |
| VILA1.5 w/FastV | 25.0% | 8.2 | 100.3 |
| **VILA1.5 w/Ours** | 25.0% | **6.3** | **110.4** |

Table 14: **Efficiency analysis based on Qwen2-VL on MMVet.** We evaluate the inference costs in terms of total inference time, prefilling time, FLOPs, and KV cache memory. KV cache memory is computed with consideration of the Grouped Query Attention (GQA) used in practical inference.

| Method | Retention Ratio | FLOPs($\times$) | Total Inference Time | Prefilling Time | KV Cache | Total Speedup | Prefilling Speedup | MMVet |
|---|---|---|---|---|---|---|---|---|
| Qwen2-VL | 100% | 1.00$\times$ | 7min58s | 1min30s | 71.2MB | 1.00$\times$ | 1.00$\times$ | 52.0 |
| Qwen2-VL w/ FastV | 25% | 0.27$\times$ | 6min50s | 0min56s | 19.7MB | 1.17$\times$ | 1.61$\times$ | 33.1 |
| Qwen2-VL w/ PDrop | 25% | 0.25$\times$ | 6min49s | 0min55s | 18.1MB | 1.17$\times$ | 1.64$\times$ | 47.0 |
| Qwen2-VL w/ Dart | 25% | 0.30$\times$ | 6min51s | 0min57s | 21.6MB | 1.16$\times$ | 1.58$\times$ | 44.5 |
| **Qwen2-VL w/ Ours** | 25% | **0.24$\times$** | **6min50s** | **0min54s** | **17.8MB** | **1.17$\times$** | **1.67$\times$** | **50.8** |

