# OpenReview forum: "Task-Related Token Compression in Multimodal Large Language Models from an Explainability Perspective"
_ICLR.cc/2026/Conference — ICLR 2026 Poster_

### Official Review · Reviewer_66M8 · 2025-10-26

**Soundness:** 3
**Presentation:** 2
**Contribution:** 2
**Rating:** 6
**Confidence:** 3

**Summary:**

This paper proposes a visual token compression method for Multimodal Large Language Models (MLLMs) based on explainability approaches, enabling task-related token compression at the LLM input stage. The core innovation lies in using explainability methods to evaluate the importance of visual tokens and training a lightweight network to predict these importance scores. Evaluations on 3 MLLM models and 11 image/video benchmarks show that the method maintains high performance while reducing computational complexity, prefilling time, and KV-cache usage.

**Strengths:**

1. It achieves an innovative paradigm shift, as the study challenges the previous view that shallow-layer visual tokens are indispensable.
2. The authors conducted comprehensive experiments to demonstrate the superiority of their method compared to other existing approaches.

**Weaknesses:**

Requires more discussion and ablation analysis; see Questions for details.

**Questions:**

1. Why is the network depth set to 5 layers? No ablation analysis is provided to justify this choice.
2. Can relevance prediction be improved using more advanced architectures without sacrificing efficiency?
3. The motivation explanation in Section 3.1 is insufficiently clear. Specifically, the causal link indicated by "therefore" in Lines 186–187 is not clear enough.
4. Why is one layer of attention sufficient? The paper mentions "first-layer attention suffices" but provides no theoretical or empirical analysis to support this claim.

---

> ### Author Response · Authors · 2025-11-24
>
> We appreciate reviewer 66M8’s efforts in reviewing our paper and the valuable feedback provided.
> > Q1: Why is the network depth set to 5 layers? No ablation analysis is provided to justify this choice.
>
> We agree with the reviewer that an ablation study on network depth is necessary to justify our design choice. In response, we have conducted controlled experiments based on Qwen2-VL with varying depths (3, 5, 7, and 10 layers) using the same training data, input representation $A^0$, and training settings (learning rate, batch size, epochs). All variants follow a consistent channel growth strategy: we start with 32 channels and gradually increase capacity, with intermediate channel dimensions repeated to moderate growth and avoid over-parameterization. The exact configurations are:
> - 3-layer: [32, 64, 128]
> - 5-layer: [32, 64, 128, 256, 512]
> - 7-layer: [32, 64, 128, 128, 256, 256, 512]
> - 10-layer: [32, 64, 128, 128, 256, 256, 512, 512, 512, 512]
>
> |   |   |   |   |   |   |   |   |   |   |
> |---|---|---|---|---|---|---|---|---|---|
> |**Methods**|**Retention Ratio**|**FLOPs**|**MME**|**MMStar**|**MMVet**|**Video-MME**|**MVBench**|**MMBecnh-V**|**Avg.**|
> |Vanilla|100%|1.00x|2295.1|60.4|54.0|50.4|51.0|1.23|100|
> |3-layer|50%|0.49x|2265.0|55.5|50.4|49.8|49.3|1.15|95.5|
> |5-layer|50%|0.49x|2288.3|55.9|51.9|50.0|49.8|1.18|96.9|
> |7-layer|50%|0.49x|2279.4|56.1|51.9|49.8|49.4|1.17|96.5|
> |10-layer|50%|0.49x|2275.1|55.1|51.5|50.0|49.7|1.15|96.0|
>
> |   |   |   |   |   |   |   |   |   |   |
> |---|---|---|---|---|---|---|---|---|---|
> |**Methods**|**Retention Ratio**|**FLOPs**|**MME**|**MMStar**|**MMVet**|**Video-MME**|**MVBench**|**MMBecnh-V**|**Avg.**|
> |Vanilla|100%|1.00x|2295.1|60.4|54.0|50.4|51.0|1.23|100|
> |3-layer|25%|0.24x|2262.8|51.1|47.3|47.6|46.0|1.09|90.7|
> |5-layer|25%|0.24x|2280.9|51.8|47.3|48.1|46.7|1.11|91.7|
> |7-layer|25%|0.24x|2282.9|51.4|46.7|48.1|46.3|1.10|91.1|
> |10-layer|25%|0.24x|2273.2|51.3|45.8|47.0|46.3|1.10|90.4|
>
> The results show that the 5-layer model achieves the best performance under retention ratios of 50% and 25%. Increasing the depth to 7 or 10 layers leads to a slight performance degradation—especially under the more challenging 25% retention condition. This suggests that deeper networks begin to overfit or suffer from optimization difficulties given the simplicity of the task. In contrast, the 3-layer model lacks sufficient capacity to capture the relevance distribution. Therefore, the 5-layer architecture strikes the optimal balance between model capacity and task complexity, which justifies our design choice.
>
> > Q2: Can relevance prediction be improved using more advanced architectures without sacrificing efficiency?
>
> This point highlights a promising direction for future improvement rather than the core focus of our current work, and we agree it warrants further exploration. Our primary objective in this study is efficiency—particularly at inference time. We find it noteworthy that even a simple convolutional network can achieve the observed performance, which motivated us to release this lightweight approach to the community. In fact, more advanced architectures inherently incur additional inference overhead. Under our training-free setting, adopting complex models would undermine the very efficiency we aim to preserve.
> As discussed in Section 5 (Future Work), integrating advanced networks could be meaningful primarily in a training-based paradigm: the additional cost incurred by more sophisticated designs may be offset by substantially higher token compression ratios, which lead to shorter input sequences. In such scenarios, the overall inference efficiency could still be preserved—or even improved—despite the use of more complex components.
>
> > Q3: The motivation explanation in Section 3.1 is insufficiently clear.
>
> First, we apologize for the unclear statements. The intended meaning here is that many advanced models—such as Qwen2VL and VILA1.5—have **already** incorporated built-in compression mechanisms (e.g., spatial and temporal pooling) to reduce task-agnostic redundancy. **Therefore**, our work does not perform task-agnostic compression again, but focuses on further mitigating task-related redundancy —specifically, selecting visual tokens that carry the most task-relevant information with respect to the instruction (i.e., the task), which is the key distinction from task-agnostic compression.

---

> ### Author Response · Authors · 2025-11-24
> **Official Comment by Authors Part 2**
>
> > Q4: Why is one layer of attention sufficient? The paper mentions "first-layer attention suffices" but provides no theoretical or empirical analysis to support this claim.
>
> Thanks for the comment. We add extra experiments for this concern. We trained our lightweight model based on Qwen2-VL using attention map from layer 2, 4, 6 as input. Then we evaluated the variants on three image benchmarks and three video benchmarks (using the same evaluation framework and FLOPs computation as in the manuscript). The results are as follows.
>
> |   |   |   |   |   |   |   |   |   |   |
> |---|---|---|---|---|---|---|---|---|---|
> |**Methods**|**Retention Ratio**|**FLOPs**|**MME**|**MMStar**|**MMVet**|**Video-MME**|**MVBench**|**MMBecnh-V**|**Avg.**|
> |Vanilla|100%|1.00x|2295.1|60.4|54.0|50.4|51.0|1.23|100|
> |1-layer|50%|0.49x|2288.3|55.9|51.9|50.0|49.8|1.18|96.9|
> |2-layer|50%|0.53x|2279.8|55.8|52.0|50.1|50.2|1.18|97.0|
> |4-layer|50%|0.60x|2303.0|56.3|52.2|49.3|50.1|1.15|96.6|
> |6-layer|50%|0.67x|2278.1|56.7|49.5|49.4|49.6|1.19|96.1|
>
> |   |   |   |   |   |   |   |   |   |   |
> |---|---|---|---|---|---|---|---|---|---|
> |**Methods**|**Retention Ratio**|**FLOPs**|**MME**|**MMStar**|**MMVet**|**Video-MME**|**MVBench**|**MMBecnh-V**|**Avg.**|
> |Vanilla|100%|1.00x|2295.1|60.4|54.0|50.4|51.0|1.23|100|
> |1-layer|25%|0.24x|2280.9|51.8|47.3|48.1|46.7|1.11|91.7|
> |2-layer|25%|0.28x|2272.0|51.9|46.6|47.8|48.4|1.10|91.7|
> |4-layer|25%|0.35x|2270.2|52.1|48.0|48.0|48.1|1.11|92.3|
> |6-layer|25%|0.43x|2252.3|52.1|47.2|47.4|47.6|1.11|91.6|
>
> The results show that using multi-layer attention scores as input yields only marginal gains—+0.1% average at 50% retention (with 2 layers) and +0.6% at 25% retention (with 4 layers)—yet incurs substantial computational overhead, with FLOPs increasing from 0.24× to 0.43× at 25% retention, almost two times. Notably, the first-layer input achieves nearly the same performance as multi-layer variants while being significantly more efficient.
> Our work aims to achieve an innovative paradigm shift that enables task-related token compression to be applied prior to the LLM, as you noted in the strengths, and we appreciate your recognition. Compressing visual tokens prior to LLM input offers notable advantages: it significantly reduces computation and memory overhead during both prefill and decode phases, and its model-agnostic nature allows deployment without modifying the LLM architecture. However, leveraging deeper-layer attention maps contradicts this design goal. Overall, the first-layer attention map suffices for our purpose.
>
> Following Reviewer 66M8's suggestion, we will revise unclear statements and include additional ablation studies to ensure the completeness of our experimental evaluation.

---

### Official Review · Reviewer_y6e3 · 2025-10-30

**Soundness:** 3
**Presentation:** 3
**Contribution:** 3
**Rating:** 4
**Confidence:** 4

**Summary:**

This paper aims to compress visual tokens for accelerating inference of multi-modal large language models (MLLMs).

Specifically, it transfers the explainability methods for transformer-based architectures to visual token pruning.
The explainability methods often keep a relevance map that could be used to measure the importance of visual tokens.

To validate the effectiveness of the method, experiments on various benchmarks (MME, MMStar, MMVet, Video-MME, MVbench, and MMBench-V) are conducted, demonstrating improvements over previous methods.

**Strengths:**

(1) The paper writes clearly and is easy to follow.

(2) It is interesting that the relevance map from explainability methods can accurately identify important visual tokens.

**Weaknesses:**

(1) To derive the relevance map, it requires the ground truth labels to calculate gradients. Thus, the paper proposes a lightweight model to distill knowledge from the derived relevance maps. However, it is challenging to make it generalizable to various data, as the lightweight model is small and the size of training data is also much smaller than that used for MLLMs.

It would be better to evaluate the model performance on various benchmarks, like GQA, SQA, VQAv2, VizWiz, and MMB.


(2) For the lightweight model, it takes the first-layer attention scores as inputs.
     Why not use the original visual tokens as inputs?
     Is it helpful to use multiple-layer attention scores?

**Questions:**

See weaknesses.

---

> ### Author Response · Authors · 2025-11-24
>
> We appreciate your recognition of our idea of “compressing visual tokens from an explanation perspective.” We hope the following rebuttal helps address your concerns.
> > Q1: It is challenging to make it generalizable to various data, as the lightweight model is small and the size of training data is also much smaller than that used for MLLMs.
>
> We understand Reviewer y6e3’s concern regarding generalization due to the relatively limited scale of training. We would like to address this concern in the following three aspects:
> First, our method demonstrates strong performance—particularly noteworthy is the fact that the training set and the test benchmark are different. Moreover, we included generalization experiments in Section 4.3 'Applying to Larger Images and Longer Videos', where we evaluate the model’s ability to generalize to inference inputs with far more tokens than seen during training.
> Second, we would like to theoretically analyse that learning such a mapping may not be as difficult as it seems.
> - **The convolutional mapper learns a structurally matched input–output mapping.** As shown in the relevance propagation equation, the relevance map $R_t$ is essentially an iterative composition of attention maps $A_t^l$ across layers.  Therefore, using $A$ to predict $R$ is natural and intuitively well-motivated.
> $$R_t = R_t + \mathbb{E}_h((A_t^l \odot \nabla A_t^l)^+)\cdot R_t$$
>   Here, we apologize for the omission of the plus sign in the relevance propagation equation (Eq.1 in Section 3.2) in our manuscript.
> - **Shallow-layer attention maps contain vital information.**  It has been demonstrated in several works [1,2] that visual tokens make a greater contribution to output generation in the shallow layers compared to the deeper layers. Accordingly, to trade off effectiveness and efficiency, we use the first-layer attention map as input.
> - **We aim not to learn an identical $R_t$ , but rather its relatively large values.** The loss function design in Section 3.3 masked the bottom 50% of label values, simplifying the learning task to the distribution of the top 50%. Our goal is not to learn a precise one-to-one input–output mapping. Instead, the learning target is only to achieve identifying which regions are relevant to the instruction, thereby enabling token pruning. One can observe the visualization results in Appendix A.1 for evidence. The pruning results $\hat{V}$ (based on $R_v$) and the convolutional network’s learned $\hat{\tilde{V}}$ (based on $\tilde{R_v}$) are not identical, yet it retains the same task-related regions and produces correct answers.
>
> Third, we have conducted additional experiments on more benchmarks, which further validate the effectiveness and generalization of our approach. Please refer to our response to Q2 and to Reviewer M6Sq’s Q2 for further information.
> We also find this to be an interesting observation and leave it open to the community, hoping it may offer useful insights.
>
> [1] Chen, Liang, et al. "An image is worth 1/2 tokens after layer 2: Plug-and-play inference acceleration for large vision-language models." European Conference on Computer Vision. Cham: Springer Nature Switzerland, 2024.
> [2] Xing, Long, et al. "Pyramiddrop: Accelerating your large vision-language models via pyramid visual redundancy reduction." arXiv preprint arXiv:2410.17247 (2024).

---

> ### Author Response · Authors · 2025-11-24
> **Official Comment by Authors Part 2**
>
> > Q2: It would be better to evaluate the model performance on various benchmarks.
>
> To address this concern, we conducted experiments on the benchmarks mentioned based on Qwen2-VL, and additionally included FastV and DART as baselines (using the same configurations and evaluation framework as in the manuscript). The results are as follows. Due to time constraints, we used the VQAv2 validation set with available answers for evaluation. The validation set contains over 200k samples, which was too large to run all evaluations during the rebuttal period. We therefore randomly sampled 50 questions from each of the 65 question types, resulting in a subset of 3,250 samples from validation set. In later versions, we will evaluate our method on the full VQAv2 test set.
> |   |   |   |   |   |   |   |   |
> |---|---|---|---|---|---|---|---|
> |**Methods**|**Retention Ratio**|**FLOPs**|**GQA**|**SQA**|**VQAv2**|**VizWiz**|**MMB**|
> |Qwen2-VL|50%|1.00X|62.2|85.7|80.1|44.3|81.0|
> |Qwen2-VL w/ FastV|50%|0.51X|43.9|65.3|67.9|32.5|42.1|
> |Qwen2-VL w/ Dart|50%|0.51X|61.3|84.7|79.0|43.9|80.1|
> |Qwen2-VL w/ Ours|50%|0.49X|**61.6**|**85.5**|**79.1**|**44.1**|**80.9**|
>
> |   |   |   |   |   |   |   |   |
> |---|---|---|---|---|---|---|---|
> |**Methods**|**Retention Ratio**|**FLOPs**|**GQA**|**SQA**|**VQAv2**|**VizWiz**|**MMB**|
> |Qwen2-VL|25%|1.00X|62.2|85.7|80.1|44.3|81.0|
> |Qwen2-VL w/ FastV|25%|0.27X|41.5|64.3|64.7|31.5|39.1|
> |Qwen2-VL w/ Dart|25%|0.27X|58.8|83.9|**77.1**|41.3|77.5|
> |Qwen2-VL w/ Ours|25%|0.24X|**59.3**|**84.3**|76.8|**43.1**|**77.7**|
>
> As shown in the results, our method demonstrates strong performance across new benchmarks, consistently retaining 95% of its original performance even at a 25% token retention rate.

---

> > ### Comment · Reviewer_y6e3 · 2025-11-27
> > **Interesting results**
> >
> > Thank you to the authors for their detailed feedback.
> >
> > Although I still do not fully understand how such a small model trained on limited data can generalize to open-world data at inference,  the experimental results appear to support this capability.
> >
> > Thus, I would like to raise my score to 6.

---

> > > ### Author Response · Authors · 2025-11-29
> > >
> > > Dear Reviewer y6e3,
> > >
> > > We sincerely appreciate your recognition, and we’re glad that our clarification addressed your concerns and that you found the results interesting. Thanks again for taking the time to raise the rating.
> > >
> > > Best regards,
> > > Authors

---

> ### Author Response · Authors · 2025-11-24
> **Official Comment by Authors Part 3**
>
> > Q3(1):  Why not use the original visual tokens as inputs?
>
> The choice of input for the lightweight convolutional model is mainly based on formulation of the relevance score $R_t$, i.e.,
> $$R_t = R_t + \mathbb{E}_h((A_t^l \odot \nabla A_t^l)^+)\cdot R_t$$
> As we stated before, essentially, $R_t$is obtained by aggregating attention maps. Therefore, we consider learning a mapping from attention maps to $R_t$ to be a promising approach. Moreover, visual tokens contribute more to output generation in shallow layers than in deeper ones [1,2], indicating that attention maps from shallow layers preserve more task-relevant information. Accordingly, to trade off effectiveness and efficiency, we use the first-layer attention map$A^0$as input.
> We do not use the original visual token embeddings as inputs because the learning target is the global relevance scores of visual tokens. Directly learning a mapping from the visual token embeddings to a global relevance score for each token is challenging, because they lack a clear, structured correspondence—like the $A \to R_t$relationship we analyzed in the previous paragraph.
>
> [1] Chen, Liang, et al. "An image is worth 1/2 tokens after layer 2: Plug-and-play inference acceleration for large vision-language models." European Conference on Computer Vision. Cham: Springer Nature Switzerland, 2024.
> [2] Xing, Long, et al. "Pyramiddrop: Accelerating your large vision-language models via pyramid visual redundancy reduction." arXiv preprint arXiv:2410.17247 (2024).
> > Q3(2): Is it helpful to use multiple-layer attention scores?
>
> We appreciate the reviewer’s insightful consideration regarding the use of multi-layer attention maps as input. To investigate this, we trained our lightweight model based on Qwen2-VL using attention map from layer 2, 4, 6 as input. We evaluated the variants on three image benchmarks and three video benchmarks (using the same evaluation framework and FLOPs computation as in the manuscript). The results are as follows.
> |   |   |   |   |   |   |   |   |   |   |
> |---|---|---|---|---|---|---|---|---|---|
> |**Methods**|**Retention Ratio**|**FLOPs**|**MME**|**MMStar**|**MMVet**|**Video-MME**|**MVBench**|**MMBecnh-V**|**Avg.**|
> |Vanilla|100%|1.00x|2295.1|60.4|54.0|50.4|51.0|1.23|100|
> |1-layer|50%|0.49x|2288.3|55.9|51.9|50.0|49.8|1.18|96.9|
> |2-layer|50%|0.53x|2279.8|55.8|52.0|50.1|50.2|1.18|97.0|
> |4-layer|50%|0.60x|2303.0|56.3|52.2|49.3|50.1|1.15|96.6|
> |6-layer|50%|0.67x|2278.1|56.7|49.5|49.4|49.6|1.19|96.1|
>
> |   |   |   |   |   |   |   |   |   |   |
> |---|---|---|---|---|---|---|---|---|---|
> |**Methods**|**Retention Ratio**|**FLOPs**|**MME**|**MMStar**|**MMVet**|**Video-MME**|**MVBench**|**MMBecnh-V**|**Avg.**|
> |Vanilla|100%|1.00x|2295.1|60.4|54.0|50.4|51.0|1.23|100|
> |1-layer|25%|0.24x|2280.9|51.8|47.3|48.1|46.7|1.11|91.7|
> |2-layer|25%|0.28x|2272.0|51.9|46.6|47.8|48.4|1.10|91.7|
> |4-layer|25%|0.35x|2270.2|52.1|48.0|48.0|48.1|1.11|92.3|
> |6-layer|25%|0.43x|2252.3|52.1|47.2|47.4|47.6|1.11|91.6|
>
> The results show that using multiple layers attention scores offers some help (i.e., +0.1% average at 50% retention using 2 layers and +0.6% average at 25% retention using 4 layers). However, this benefit comes with substantial computational overhead (e.g., from 0.24× to 0.43× FLOPs at 25% retention, almost two times).
> Such increased computation and latency run counter to the core objective of our work, we prioritize inference efficiency and opt for the first-layer attention map as a principled trade-off between effectiveness and efficiency. More importantly, our approach provides concrete evidence that effective task-oriented visual token compression can be successfully performed prior to the LLM—the use of multiple layers attention maps undermines this very goal.
>
> Following reviewer y6e3's suggestion, we will include the mentioned benchmarks in the experimental section, enrich the analyses with more detailed discussions, and supplement additional ablation studies to further justify the soundness of our design.

---

### Official Review · Reviewer_M6Sq · 2025-10-31

**Soundness:** 3
**Presentation:** 3
**Contribution:** 3
**Rating:** 6
**Confidence:** 4

**Summary:**

The paper proposes a task-related token compression paradigm for Multimodal Large Language Models (MLLMs) to address the high computational cost and inefficiency associated with a large number of visual tokens in the LLM input. The authors first use an explainability method (e.g., gradient-weighted multi-head attention) to assess the global importance of each visual token relative to a given instruction, generating a "ground truth" importance score. Building on this insight, they train a lightweight convolutional network to predict this importance score based solely on the LLM's first layer attention map. Experiments demonstrate the effectiveness of the proposed method.

**Strengths:**

1. The paper is well-written and easy to understand. Figure 1 and 2 are intuitive to understand the motivation and the big picture of the proposed method.

2. The authors successfully demonstrate the feasibility and efficiency of performing task-related token compression at the LLM input stage with negligible performance loss, providing a new, more efficient compression strategy.

3. Experiments validate the effectiveness across three distinct MLLM architectures (Qwen2-VL, LLaVA-One Vision, VILA1.5) and 11 diverse image and video benchmarks, showcasing its robustness and wide applicability.

**Weaknesses:**

1. The lightweight convolutional network is trained using only the first-layer attention map ($A^0$). A more in-depth analysis is needed to justify how such a small network, using only shallow information, can accurately and robustly predict the global, task-specific importance score, which typically requires information propagated through multiple LLM layers.

2. This train-based methods may overfit to the training data. I wonder the performance on some OCR related benchmarks (TextVQA, Chartqa, DocVQA, OCRBench), since they are more challenging to validate the effectiveness of the proposed method.

**Questions:**

The paper mentions training using only 10K image samples. Please elaborate on the data sampling strategy (e.g., are they sampled from the test benchmarks? Are they general-domain? How is diversity ensured?) Is the small scale of the training data a potential limitation when generalizing to open-world or other benchmarks scenarios beyond the 11 evaluation benchmarks?

---

> ### Author Response · Authors · 2025-11-24
>
> We appreciate reviewer M6Sq’s efforts in reviewing our paper and the valuable feedback provided.
> > Q3: Please elaborate on the data sampling strategy (e.g., are they sampled from the test benchmarks? Are they general-domain? How is diversity ensured?) Is the small scale of the training data a potential limitation when generalizing to open-world or other benchmarks scenarios beyond the 11 evaluation benchmarks?
>
> We train our explainability-based compressor based on subsets sampled from general-domain high-quality datasets Infinity-MM and LLaVAVideo. These open-source datasets and the test benchmark are different. To ensure high diversity, we adopt a sampling strategy that covers a wide range of task types and video duration. The details of the data sampling strategy are as follows (Actually we have included it in Appendix B, please refer to them.):
> - Image Dataset. For training the compressor used in image tasks, we sample a subset of Infinity-MM that ensures high quality and diversity. The training set primarily consists of data used during Stage 4, including 9k samples randomly sampled from the Data Generated by GPT-4 subset and 4k from Synthetic Data.
> - Video Dataset. For training the compressor used in video tasks, we sample a subset of LLaVAVideo. Specifically, we include 7k samples from LLaVA-Video, 6k from NeXT-QA and 4k from ActivityNetQA. Note that the training sets of NeXT-QA and ActivityNet-QA are different from the test sets used in the evaluation. During sampling, since LLaVA-Video contains several parts categorized by task type (open-ended and multi-choice) and video duration (0–30s, 30-60s, 1–2min and 2-3min), we ensure a balanced distribution by randomly selecting an equal number of training examples from each part.
>
> We use a relatively small amount of training data primarily because we employ a small model to ensure inference efficiency. Nevertheless, we understand the concern regarding generalization due to the relatively limited scale of training. We would like to address this concern in the following three aspects:
> 1. Our method demonstrates strong performance—particularly noteworthy is the fact that the training set and the test benchmark are different. Moreover, we included generalization experiments in Section 4.3 'Applying to Larger Images and Longer Videos', where we evaluate the model’s ability  to generalize to inference inputs with far more tokens than seen during training.
> 2. We would like to theoretically analyse that learning such a mapping may not be as difficult as it seems. Please refer to our response to Q1 for details.
> 3. Following your suggestion and that of Reviewer y6e3, we have conducted additional experiments on more benchmarks, which further validate the effectiveness and generalization of our approach. Please refer to our response to Q2 and to Reviewer y6e3’s Q2 for further information.
>
> We also find this to be an interesting observation and leave it open to the community, hoping it may offer useful insights.

---

> ### Author Response · Authors · 2025-11-24
> **Official Comment by Authors Part 2**
>
> > Q1: A more in-depth analysis is needed to justify how such a small network, using only shallow information, can accurately and robustly predict the global, task-specific importance score, which typically requires information propagated through multiple LLM layers.
>
> We will conduct a more in-depth analysis from both theoretical and practical perspectives.
> 1. **The convolutional mapper learns a structurally matched input–output mapping.** As shown in the relevance propagation equation, the relevance map $R_t$ is essentially an iterative composition of attention maps $A_t^l$ across layers.  Therefore, using $A$ to predict $R$ is natural and intuitively well-motivated.
> $$R_t = R_t + \mathbb{E}_h((A_t^l \odot \nabla A_t^l)^+)\cdot R_t$$
> Here, we apologize for the omission of the plus sign in the relevance propagation equation (Eq.1 in Section 3.2) in our manuscript.
> 2. **Shallow-layer attention maps contain vital information.** It has been demonstrated in several works [1,2] that visual tokens make a greater contribution to output generation in the shallow layers compared to the deeper layers. Accordingly, to trade off effectiveness and efficiency, we use the first-layer attention map as input.
> 3. **We aim not to learn an identical $R_t$ , but rather its relatively large values.** The loss function design in Section 3.3 masked the bottom 50% of label values, simplifying the learning task to the distribution of the top 50%. Our goal is not to learn a precise one-to-one input–output mapping. Instead, the learning target is only to achieve identifying which regions are relevant to the instruction, thereby enabling token pruning. One can observe the visualization results in Appendix A.1 for evidence. The pruning results $\hat{V}$ (based on $R_v$) and the convolutional network’s learned $\hat{\tilde{V}}$ (based on $\tilde{R_v}$) are not identical, yet it retains the same task-related regions and produces correct answers.
> 4. **Extensive experiments validate its effectiveness and generalization.** We evaluate our approach on 11 image and video benchmarks with three leading MLLMs and show its strong performance. Moreover, we apply our model directly to images and videos that are larger and longer ('Applying to Larger Images and Longer Videos' in Section 4.3), than those used in training, and consistently observe competitive performance, highlighting its robustness and generalization capability. Additional experiments presented in our response to Q2 and to Reviewer y6e3’s Q2 provide further validation.
>
> [1] Chen, Liang, et al. "An image is worth 1/2 tokens after layer 2: Plug-and-play inference acceleration for large vision-language models." European Conference on Computer Vision. Cham: Springer Nature Switzerland, 2024.
> [2] Xing, Long, et al. "Pyramiddrop: Accelerating your large vision-language models via pyramid visual redundancy reduction." arXiv preprint arXiv:2410.17247 (2024).
>
> > Q2: OCR related benchmarks are more challenging to validate the effectiveness of the proposed method.
>
> We fully agree with the reviewer M6Sq that OCR-related benchmarks are challenging for visual token compression. To address this concern, we conducted experiments on the OCR benchmarks mentioned based on Qwen2-VL, and additionally included FastV and DART as baselines (using the same configurations and evaluation framework as in the manuscript). The results are as follows.
> |   |   |   |   |   |   |   |
> |---|---|---|---|---|---|---|
> |**Methods**|**Retention Ratio**|**FLOPs**|**TextVQA**|**Chartqa**|**DocVQA**|**OCRBench**|
> |Qwen2-VL|50%|1.00X|84.1|92.7|93.1|81.5|
> |Qwen2-VL w/ FastV|50%|0.51X|66.7|51.3|59.2|56.3|
> |Qwen2-VL w/ Dart|50%|0.51X|**82.7**|78.4|79.7|78.5|
> |Qwen2-VL w/ Ours|50%|0.49X|82.6|**80.1**|**81.2**|**79.5**|
> |   |   |   |   |   |   |   |
>
> |   |   |   |   |   |   |   |
> |---|---|---|---|---|---|---|
> |**Methods**|**Retention Ratio**|**FLOPs**|**TextVQA**|**Chartqa**|**DocVQA**|**OCRBench**|
> |Qwen2-VL|25%|1.00X|84.1|92.7|93.1|81.5|
> |Qwen2-VL w/ FastV|25%|0.27X|63.2|41.1|46.1|46.8|
> |Qwen2-VL w/ Dart|25%|0.27X|76.7|65.0|62.8|65.1|
> |Qwen2-VL w/ Ours|25%|0.24X|**80.0**|**70.9**|**72.1**|**67.8**|
> |   |   |   |   |   |   |   |
>
> As shown in the results, our method achieves strong performance on OCR-related benchmarks, with particularly pronounced gains under lower visual token retention rates.
> Following reviewer M6Sq's suggestion, we will provide more detailed implementation specifics (such as data sources and sampling strategy) in the main text, include the mentioned benchmarks in the experimental section, and conduct a more in-depth analysis of the experiments results.

---

### Official Review · Reviewer_RQ2d · 2025-11-06

**Soundness:** 3
**Presentation:** 3
**Contribution:** 3
**Rating:** 6
**Confidence:** 4

**Summary:**

This paper introduces a novel approach for task-aware visual token compression in multimodal large language models (MLLMs), aiming to eliminate instruction-irrelevant tokens at the LLM input stage to enhance inference efficiency—without modifying the underlying architecture. The key idea is to utilize explainability techniques to assign relevance scores to visual tokens with respect to a given instruction, guiding the compression process accordingly. Furthermore, the authors demonstrate that a lightweight convolutional network can be trained to map first-layer attention maps to explainability-derived importance scores, enabling token importance prediction without a full forward pass.

**Strengths:**

1. The work tackles a critical bottleneck in MLLMs that high computational and memory overhead from large numbers of visual tokens by introducing an instruction-aware token compression paradigm that operates without architectural changes.
2. The proposed compressors are lightweight, transferable, and require minimal retraining across diverse MLLM architectures, underscoring strong generalizability.
3. Evaluation spans 11 benchmarks for both images and videos, with comprehensive ablations, efficiency analyses, and baseline comparisons.

**Weaknesses:**

1. The paper relies heavily on empirical findings and visual evidence, offering limited theoretical grounding. The mathematical formulation (e.g., the relevance propagation equation for 𝑅t in Section 3.2) lacks a deeper analysis of properties such as linearity, gradient behavior, and attribution faithfulness, especially under ambiguous or multi-factor instructions.

**Questions:**

Please refer to weakness.

---

> ### Author Response · Authors · 2025-11-24
>
> We appreciate reviewer RQ2d’s efforts in reviewing our paper and the valuable feedback provided.
> The explainability method used in our paper follow the widely used attention-based explanation framework GAE [1] and its predecessor [2]. These two works offer comprehensive derivations and partial theoretical guarantees for attention-based explainability methods, and further delineate the methodological transition from LRP-based relevance[2] to attention-based relevance[1]. Attention-based[1, 4] explanations assume that attention reflects meaningful intermediate correlations for prediction[5]. Overall, their underlying principles and the assumptions adopted by many explainability methods are aligned—namely, that larger weights indicate greater importance of the corresponding factors, i.e., tokens in Transformer architectures.
> The properties that reviewer RQ2d highlighted are highly insightful. Below, we analyse them based on theoretical and practical considerations. We first apologize for the omission of the plus sign in the relevance propagation equation (Eq.1 in Section 3.2) in our manuscript.
> $$R_t = R_t + \mathbb{E}_h((A_t^l \odot \nabla A_t^l)^+)\cdot R_t$$
> - **Linearity**: As noted in prior work [1,4], explainability methods tailored to Transformer architectures often prioritize alignment with the model’s intrinsic mechanisms (e.g., attention patterns) over adherence to abstract axiomatic properties like linearity. In this context, despite its nonlinearity, we argue that GAE arises naturally from leveraging the Transformer’s native attention dynamics for explanation.  Moreover,  the broader explainability literature increasingly recognizes that no single set of theoretical criteria (e.g., linearity, sensitivity) universally guarantees explainability. Practical utility, stability, and coherence with human intuition are regarded as equally important dimensions of effective explanation [5].
> - **Gradient behavior**: As shown in the equation, the relevance information of each layer is computed via the Hadamard product of the attention map and its gradient. The attention map captures how much attention each token receives from other tokens, while the gradient identifies which tokens require more attention to effectively influence the output.  Following common practice in gradient-based attribution, GAE zeros out negative gradient components. In our view, this rectification not only enforces a causal direction of influence but also prevents the suppression of informative positive signals by accumulated negative components [2, 4]. Additionally, incorporating gradient addresses the issue highlighted in GAE [1] that simple averaging across attention heads can lead to distorted aggregation of head-specific information. We empirically validate the effectiveness of this gradient-aware aggregation through comparative experiments in Section 4.4.
> - **Attribution faithfulness**：While our work does not aim to propose a new explainability method—instead leveraging the established attention-gradient framework from GAE [1] to address visual redundancy—we acknowledge the importance of attribution faithfulness. Faithfulness, in this context, refers to whether the computed relevance scores (i.e., $R_t$) genuinely reflect each visual token’s contribution to the model’s output. Although we do not include dedicated faithfulness metrics in our main experiments, several lines of evidence support the reliability of the attributions.
>   First, $R_t$ is derived from both the raw attention map and its task-specific gradient, ensuring that relevance reflects not only structural connectivity but also functional sensitivity to the current instruction. In Appendix A.2, we provide a case study demonstrating that $R_t$ adaptively highlights different visual regions under diverse instructions—from high-level summarization to fine-grained queries about clothing—demonstrating semantic alignment with task intent.
>   Second, our token pruning strategy implicitly validates faithfulness: by removing visual tokens with low relevance scores and observing minimal performance degradation (across 11 benchmarks), we effectively perform a demonstration akin to the masking-based faithfulness evaluation in prior work [6]. If $R_t$ were unfaithful (e.g., ranking irrelevant tokens as important), such pruning would lead to significant performance drops, which we do not observe.
>   Finally, GAE [1]—the explainaiton method we adopted—has already been evaluated for faithfulness under similar protocols. In this light, our results can be viewed as an indirect but extensive empirical confirmation of the method’s faithfulness in multimodal reasoning scenarios.

---

> ### Author Response · Authors · 2025-11-24
> **Official Comment by Authors Part 2**
>
> Following reviewer RQ2d's suggestion, we will further strengthen both the theoretical analysis and the evaluation of attribution faithfulness. In the revised version, we will supplement the appendix with a section that offers a detailed interpretation of the relevance propagation equation and more comprehensive case studies for assessing faithfulness.
> [1] Chefer, Hila, Shir Gur, and Lior Wolf. "Generic attention-model explainability for interpreting bi-modal and encoder-decoder transformers." Proceedings of the IEEE/CVF international conference on computer vision. 2021.
> [2] Chefer, Hila, Shir Gur, and Lior Wolf. "Transformer interpretability beyond attention visualization." Proceedings of the IEEE/CVF conference on computer vision and pattern recognition. 2021.
> [3] Bach, Sebastian, et al. "On pixel-wise explanations for non-linear classifier decisions by layer-wise relevance propagation." PloS one 10.7 (2015): e0130140.
> [4] Barkan, Oren, et al. "Grad-sam: Explaining transformers via gradient self-attention maps." Proceedings of the 30th ACM International Conference on Information & Knowledge Management. 2021.
> [5] Zhao, Haiyan, et al. "Explainability for large language models: A survey." ACM Transactions on Intelligent Systems and Technology 15.2 (2024): 1-38.
> [6] DeYoung, Jay, et al. "ERASER: A benchmark to evaluate rationalized NLP models." Proceedings of the 58th annual meeting of the association for computational linguistics. 2020.

---

### Author Response · Authors · 2025-12-01

We sincerely thank `Reviewer RQ2d`, `Reviewer M6Sq`, `Reviewer y6e3`, and `Reviewer 66M8` for their thoughtful and constructive feedback. We are pleased that all four reviewers recognized the value of our work and offered positive assessments.

We addressed the comments in the individual responses and updated the paper accordingly, with changes highlighted in **blue**. In summary, the main revisions are:
- In Section 3.1 (line184~189), we improved the motivation explanation to make it more clear. (`66M8-Q3`)
- In Section 3.2 (Eq.1), we corrected the typographical error in relevance propagation equation.
- In Section 4.1 (line 311~317), we added more detailed implementation specifics of training $f_\theta$. (`M6Sq-Q1`)
- In Section 4.3 (Table 2),  we expanded the image task evaluation with the new benchmarks mentioned (TextVQA and MMBench) in the rebuttal. (`M6Sq-W2`, `y6e3-W1`)
- In Section 4.3 (line 470-485),  we conducted additional analysis and discussion to investigate how a lightweight $f_\theta$ can achieve strong performance (`M6Sq-W1`, `y6e3-W1`)
- In Appendix A, we added a detailed interpretation of the relevance propagation equation Eq.1. (`RQ2d-W1`)
- In Appendix E, we added additional ablation studies of different configurations for training $f_\theta$, specifically examining (i) the depth of the convolutional network and (ii) the attention layer index used as input. (`y6e3-W2`, `66M8-Q1`, `66M8-Q4`)
- In Appendix F, we expanded the image task evaluation with the new benchmarks mentioned (GQA, SQA, VizWiz, ChartQA, DocVQA and OCRBench) in the rebuttal. (`M6Sq-W2`, `y6e3-W1`)

We once again thank all the reviewers for their insightful comments, which helped improve the clarity and completeness of our paper. We hope the revised manuscript and detailed responses address all concerns.

---

### Author Response · Authors · 2025-12-03
**Summary comment by Authors**

**Dear AC**,
We regret the inconvenience caused by the unexpected OpenReview bug, and sincerely appreciate the time and effort you invested in managing the review process. To facilitate a quick understanding of the key information, we provide a summary of our paper and the rebuttal process.


**Brief Summary of Our Paper:**
Our work demonstrates that effective task-related visual token compression prior to the LLM is indeed feasible. This is regarded by `Reviewer 66M8` as "*an innovative paradigm shift*", as it "*challenges the previous view that shallow-layer visual tokens are indispensable*".
We first observe that **explainability methods** can assess the relevance of visual tokens with respect to the instruction, and the resulting importance scores can guide lossless visual compression. We further demonstrate that **a lightweight convolutional network** can be efficiently trained to map first-layer attention maps to the explainability-derived importance scores, thereby enabling task-related token compression prior to the LLM.
Extensive experiments on **13 image benchmarks** (5 from the original manuscript and 8 added during the rebuttal) and **6 video benchmarks** across **three leading MLLMs** (Qwen2-VL, LLaVA-OneVision, and VILA1.5) demonstrate the remarkable effectiveness and strong generalization of our approach.
Notably, our work offers several **key advantages**: (i) It substantially reduces computation and memory overhead during both prefill and decode phases; (ii) It does not rely on model-specific behaviors but is broadly applicable across different MLLMs, allowing deployment without modifying the MLLM architecture.


**Brief Summary of the Rebuttal Process:**
We were honored by the reviewers' **unanimous recognition of our idea**. `Reviewer RQ2d` noted that our work "*tackles a critical bottleneck in MLLMs*", `Reviewer M6Sq` highlighted that our work "*provides a new, more efficient compression strategy*", `Reviewer y6e3` found our use of explainability for token compression to be "*interesting*", and `Reviewer 66M8` described our approach as "*an innovative paradigm shift*". We were also encouraged that all reviewers **rated the soundness of our work as 3 (good)**. In addition, `Reviewer M6Sq` described the paper as "*well-written*" and remarked that the figures provide "*an intuitive understanding of the motivation*".

During the rebuttal process, we made every effort to address each constructive comment and to improve both the clarity of our presentation and the completeness of our experiments. The main revisions are summarized below:
- **Supplementary theoretical analysis.** Following `Reviewer RQ2d`'s suggestion, we strengthened the theoretical component by adding a new section in Appendix A that provides a detailed interpretation of the relevance propagation equation, including analyses of properties such as gradient behavior and attribution faithfulness.
- **Evaluation on additional image benchmarks.** Following the suggestions of `Reviewer M6Sq` and `Reviewer y6e3`, we expanded the image task evaluation by including eight new benchmarks.  The results, presented in Section 4.3 and Appendix F, further demonstrate the effectiveness and generalization of our approach.
- **Ablation studies on different configurations for training $f_\theta$.** Following the suggestions of `Reviewer y6e3` and `Reviewer 66M8`, we added additional ablation studies in Appendix E, examining different configurations for training $f_\theta$, including the effect of network depth and the use of attention scores from different layers as inputs.
- **Expanded discussion on relevance score learning.** Following the suggestions of `Reviewer M6Sq` and `Reviewer y6e3`, we conducted additional analyses and discussion in Section 4.3 for a better understanding of how a lightweight $f_\theta$ can achieve strong performance. We were delighted that Reviewer y6e3 recognized the "interesting results" during the rebuttal and raised the score.
- **Improved writing quality and organization.** Following the suggestions of `Reviewer M6Sq` and `Reviewer 66M8`, we refined the paper to improve clarity and presentation.

Despite the premature termination of the rebuttal process caused by an unexpected OpenReview bug, we were pleased that `Reviewer y6e3` had already raised his/her score from 4 to 6 before the bug occurred (recorded at 11/26 10:40:58 PM EST), expressing clear recognition of our approach. Unfortunately, we were unable to continue discussions with the other three reviewers. As a result of the rebuttal, our overall score increased from **6664 (conf: 4434)** to **6666 (conf: 4434)**.

We hope the summary will be helpful for your decision. A heartfelt thank you for your time and effort, and for your substantial contributions to the community.

Best regards,
The Authors

---

### Meta-Review · Area_Chair_z6Cv · 2025-12-28

**Summary:**

Four reviews are received on this paper.

Reviewer RQ2d has concerns on the theoretical grounding and mathematical formulation of the proposed method.

Reviewer M6Sq requested for an in-depth analysis to justify the performance of the small network, and more experiments on some OCR related benchmarks.

Reviewer y6e3 has concerns on the generalization performance of the lightweight model, and requested more experimental results on benchmarks like GQA, SQA, VQAv2, VizWiz, and MMB.

Reviewer 66M8 was overall positive on the work, and requested more discussion and ablation analysis.

**Reviewer Concerns:**

Reviewers RQ2d, M6Sq and 66M8 didn’t provide feedback on the authors’ rebuttal. The ACs read the rebuttal and believe that their concerns can be addressed since the authors presented detailed rebuttal and additional results.
While Reviewer y6e3 still has concerns on why such a small model trained on limited data can generalize to open-world data at inference, he/she was convinced by the authors’ additional experimental results and raised the score to 6.

**Reviewer Scores:**

It is clear that Reviewers RQ2d, M6Sq and 66M8 will keep their scores of 6, and Reviewer will raise his/her score to 6. This paper can be accepted by ICLR.

---

### Decision · Program_Chairs · 2026-01-26

Accept (Poster)